# MAPLE: Many-Shot Adaptive Pseudo-Labeling for In-Context Learning

**Zihan Chen** [* 1]   **Song Wang** [* 1]   **Zhen Tan** [2]   **Jundong Li** [1]   **Cong Shen** [1]

## Abstract

In-Context Learning (ICL) empowers Large Language Models (LLMs) to tackle diverse tasks by incorporating multiple input-output examples, known as demonstrations, into the input of LLMs. More recently, advancements in the expanded context windows of LLMs have led to many-shot ICL, which uses hundreds of demonstrations and outperforms few-shot ICL, which relies on fewer examples. However, this approach is often hindered by the high cost of obtaining large amounts of labeled data. To address this challenge, we propose **M**any-Shot **A**daptive **P**seudo-**L**ab**E**ling, namely **MAPLE**, a novel influence-based many-shot ICL framework that utilizes pseudo-labeled samples to compensate for the lack of label information. We first identify a subset of impactful unlabeled samples and perform pseudo-labeling on them by querying LLMs. These pseudo-labeled samples are then adaptively selected and tailored to each test query as input to improve the performance of many-shot ICL, without significant labeling costs. Extensive experiments on real-world datasets demonstrate the effectiveness of our framework, showcasing its ability to enhance LLM adaptability and performance with limited labeled data. Our code is provided at https://github.com/Chen-1031/MAPLE_ICL.

## 1. Introduction

In-Context Learning (ICL) has emerged as a fundamental capability of large language models (LLMs) (Zhao et al., 2023; Chang et al., 2024), enabling them to perform diverse tasks with a set of input-output examples, i.e., demonstrations, as input (Brown et al., 2020; Wang et al., 2023; 2024b). In contrast to fine-tuning strategies, ICL does update model parameters, making it an efficient and flexible approach for

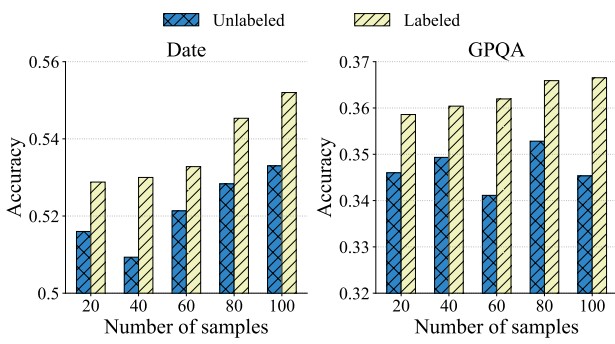

Figure 1: Accuracies on Date and GPQA datasets with different amount of demonstrations. The LLM is Gemini 1.5 Flash.

enhancing the performance of LLMs (Rubin et al., 2022; Lu et al., 2021). More recently, advancements in expanding the context windows of LLMs to accommodate a large number of input tokens have enabled *many-shot ICL* (Agarwal et al., 2024), where hundreds of demonstrations are incorporated into the input. Many-shot ICL has been shown to significantly improve performance, particularly for complex or nuanced tasks, by providing richer task-specific information (Baek et al., 2024).

However, many-shot ICL faces a critical limitation: the high cost associated with acquiring a large volume of labeled data as demonstrations (Li et al., 2024). This challenge is particularly pronounced in resource-constrained settings where manual labeling is expensive or infeasible (e.g., complex reasoning tasks), thereby limiting the applicability of many-shot ICL. An alternative strategy is to use unlabeled samples as demonstrations. However, as shown in Fig. 1, the performance improvement becomes marginal and unstable in comparison to using labeled demonstrations.

In order to effectively leverage unlabeled samples, a practical solution is to perform pseudo-labeling on them, allowing these samples to serve as demonstrations for many-shot ICL without incurring significant human labeling costs. However, this approach presents two key challenges: (1) Selection of unlabeled samples for pseudo-labeling: Given a potentially large pool of unlabeled samples, identifying the most informative samples for pseudo-labeling is a critical challenge. This is particularly relevant in tasks where unlabeled samples may contain limited information (e.g., question answering). Selecting the most beneficial unla-

---

[*]Equal contribution [1]University of Virginia, Charlottesville, VA, USA [2]Arizona State University, Tempe, AZ, USA. Correspondence to: Cong Shen <cong@virginia.edu>.

*Proceedings of the 42nd International Conference on Machine Learning*, Vancouver, Canada. PMLR 267, 2025. Copyright 2025 by the author(s).

beled samples is essential to ensure that pseudo-labeling provides meaningful additional information to enhance task performance. (2) Impact of unrelated demonstrations: With pseudo-labeled samples, the challenge remains in selecting the most suitable ones as demonstrations, tailored to each test query. Since not all pseudo-labeled and labeled samples are beneficial for inference, using unrelated demonstrations for a specific input query may involve unnecessary noise. Therefore, it is essential to adaptively select demonstrations to minimize the impact of irrelevant pseudo-labeled and labeled samples, ensuring that only the most informative and reliable ones are used as demonstrations.

In this work, we propose **M**any-Shot **A**daptive **P**seudo-**L**ab**E**ling, namely **MAPLE**, a novel influence-based framework to effectively utilize unlabeled samples for many-shot ICL. We consider the scenario where a smaller set of labeled samples and a potentially large set of unlabeled samples are available, as it is affordable to manually label several samples in practice. Our approach tackles the two primary challenges with the following designs: (1) Influence-Based Sample Selection for Pseudo-Labeling: We leverage the concept of node influence on graphs to identify the most impactful unlabeled samples relative to labeled samples. By constructing a graph encompassing both labeled and unlabeled samples, we exploit their relationships to inform selection. This ensures the pseudo-labeled samples provide the necessary information for inference. (2) Adaptive Demonstration Selection: For each input query, we adaptively select labeled and pseudo-labeled demonstrations tailored specifically to the query. By identifying and incorporating samples with the most significant influence on the test query, our approach avoids involving unrelated demonstrations and ensures the effective utilization of demonstrations. These designs collectively maximize the effectiveness of leveraging unlabeled samples, while minimizing reliance on costly labeled data and extending the applicability of LLMs to various real-world tasks with only limited labeled samples. We conduct extensive experiments on various real-world datasets, and the results validate the effectiveness of our framework. Our main contributions are summarized as follows:

- *Novelty.* We are the first to explore the capability of many-shot ICL under the pseudo-labeled setting. This novel perspective highlights the potential of leveraging abundant unlabeled samples for pseudo-labeling to alleviate the data bottleneck, broadening the applicability of ICL beyond reliance on labeled data.

- *Algorithm.* We propose an influence-based mechanism to select and pseudo-label only the most impactful unlabeled samples and adaptively select demonstrations for each test query, ensuring strong performance without extensive pseudo-labeling.

- *Practicality.* Our approach significantly reduces the need

for labeled data in many-shot ICL, thereby improving the feasibility of LLMs in real-world scenarios where labels are scarce. Through extensive experiments on diverse datasets, we demonstrate the superior performance of our framework over other baselines.

## 2. Related Works

**In-Context Learning.** In-context learning (ICL) (Brown et al., 2020) equips large language models (LLMs) with the ability to leverage a handful of input-output demonstrations for reasoning. ICL has proven remarkably successful in handling complex tasks, including summarization (Jain et al., 2023; Baek et al., 2024) and reasoning (Wang et al., 2024a; Lee et al., 2024; Chen et al., 2024b).

To effectively harness ICL, researchers have devised various adaptive strategies for selecting the most suitable demonstrations (Su et al., 2022; Chen et al., 2024a). Broadly, these methods can be grouped into learning-free and learning-based categories. The former uses heuristic strategies, including semantic similarity (Liu et al., 2021) or entropy (Lu et al., 2021), without actively querying the model (Zhao et al., 2021; Agrawal et al., 2022). Learning-based methods, in contrast, incorporate feedback from the LLM into a training loop. As a classic example, EPR (Rubin et al., 2022) applies contrastive learning and learns a score based on the probabilities of language model outputs. Moreover, CEIL (Ye et al., 2023) selects multiple demonstrations while considering their correlations, and IDS (Qin et al., 2023) iteratively select demonstrations based on zero-shot chain-of-thought reasoning (Wei et al., 2022).

**Many-Shot ICL.** With advancements in expanding the context windows of LLMs (Li et al., 2024; Bertsch et al., 2024; Team et al., 2024), many-shot ICL has emerged as a promising approach, leveraging larger sets of demonstrations to significantly boost performance at the cost of longer input lengths (Li et al., 2023; Jiang et al., 2024; Huang et al., 2024; Chen et al., 2025). As a pioneering effort, Agarwal et al. (2024) explored the capability of scaling many-shot ICL to thousands of demonstrations, showcasing its superior performance across a wide variety of tasks. While this work reduces human labeling costs by utilizing model-generated (i.e., pseudo-labeling) answers as demonstrations, it does not address the selection of unlabeled samples (for pseudo-labeling) or the adaptive selection of demonstrations for individual test queries. Baek et al. (2024) further highlight the importance of using extensive demonstrations of LLMs in many-shot settings, in contrast to the careful selection of demonstrations. However, the limited effectiveness of existing selection methods stems from their specific design for few-shot ICL (Zhang et al., 2025). This underscores the need for novel strategies tailored to the unique challenges of many-shot ICL.

# 3. Methodology

## 3.1. Preliminaries

We begin with formulating the setup of ICL in this work. Consider a dataset $\mathcal{D} = \mathcal{D}_L \cup \mathcal{D}_U$, where $\mathcal{D}_L = \{(x_i, y_i)\}_{i=1}^L$ consists of a small set of labeled samples, and $\mathcal{D}_U = \{x_j\}_{j=1}^U$ contains a large set of unlabeled samples. Here $x$ represents the textual input (i.e., query), and $y$ is the corresponding label. Generally, $L \ll U$ holds true, due to the substantial cost associated with human annotations.

The goal of (few-shot) ICL is to select a set of demonstrations $\mathcal{S} \subseteq \mathcal{D}_L$, acting as the additional input for a pre-trained language model $\mathcal{M}$. The model $\mathcal{M}$ then utilizes $\mathcal{S}$ along with a test query $x_{\text{test}}$ to make predictions:

$$\hat{y} = \mathcal{M}\left(\mathcal{S}(x_{\text{test}}), x_{\text{test}}\right), \tag{1}$$

where $\mathcal{S}(x_{\text{test}})$ denotes the set of selected demonstrations, conditioned on the test query $x_{\text{test}}$.

## 3.2. Overview

As the labeled set $\mathcal{D}_L$ may not be sufficient for many-shot ICL, we propose to identify a subset of unlabeled samples from $\mathcal{D}_U$ and perform pseudo-labeling on these samples. The overall process is described in Fig. 2. We use $\mathcal{D}_U^* \subset \mathcal{D}_U$ to denote the set of unlabeled samples selected for pseudo-labeling. The pseudo-labeling process is executed by a language model $\mathcal{M}_p$, which assigns a pseudo-label $\hat{y}_j$ to each selected sample $x_j$:

$$\hat{y}_j = \mathcal{M}_p(x_j), \quad \forall x_j \in \mathcal{D}_U^*. \tag{2}$$

Notably, $|\mathcal{D}_U^*| = P$, where the size $P$ is a hyper-parameter and serves as the pseudo-labeling budget. The pseudo-labeled samples, along with the labeled samples in $\mathcal{D}_L$, are then aggregated into $\mathcal{D}_F$, referred to as the candidate demonstration set:

$$\mathcal{D}_F = \{(x_j, \hat{y}_j) \mid x_j \in \mathcal{D}_U^*\} \cup \mathcal{D}_L. \tag{3}$$

With demonstrations in $\mathcal{D}_F$, we adaptively select a set of demonstrations $\widetilde{\mathcal{S}}(x_{\text{test}}) \subseteq \mathcal{D}_F$ for each input query $x_{test} \in \mathcal{D}_{\text{test}}$, which serves as the many-shot demonstrations for inference. The model $\mathcal{M}$ then uses these demonstrations to make predictions for $x_{\text{test}}$:

$$\hat{y} = \mathcal{M}(\widetilde{\mathcal{S}}(x_{\text{test}}), x_{\text{test}}). \tag{4}$$

## 3.3. Relationship Construction

In our framework, we aim to maximally utilize the information from the labeled set $\mathcal{D}_L$ to decide which unlabeled samples to select for pseudo-labeling. Therefore, we first construct a labeled-unlabeled graph $G = (\mathcal{V}, \mathcal{E})$ that captures the relationships among labeled and unlabeled

samples. Here $\mathcal{V}$ is the set of nodes, and $\mathcal{E}$ is the set of edges. We utilize the entire demonstration pool (i.e., dataset) $\mathcal{D} = \mathcal{D}_L \cup \mathcal{D}_U$ to build $G$, and each sample $x \in \mathcal{D}$ is treated as a node $v \in \mathcal{V}$. For each node $v_i \in \mathcal{V}$, we identify the $k$ closest nodes in terms of the relevance score $r(v_i, v_j)$, as defined in Contriver (Izacard et al., 2021), and connect them as its neighbors. The relevance score is calculated from the dot product of the encoded representations of $x_i$ and $x_j$:

$$r(v_i, v_j) = \langle f_\theta(x_i), f_\theta(x_j) \rangle, \tag{5}$$

where $x_i$ and $x_j$ are the corresponding queries of $v_i$ and $v_j$, respectively. $f_\theta(\cdot)$ is the pre-trained encoder. Notably, as the unlabeled samples in $\mathcal{D}_U$ do not contain labels (i.e., $y$), we only use the queries (i.e., $x$) to calculate the relevance score. Based on the obtained relevance scores $r(v_i, v_j)$, the adjacency matrix $\mathbf{A}$ of the labeled-unlabeled graph is obtained by connecting the $k$ nodes with the largest relevance scores to each node $v_i$. Formally, the entries of $\mathbf{A}$ are calculated as follows:

$$\mathbf{A}_{ij} = \begin{cases} r(v_i, v_j), & \text{if } r(v_i, v_j) \in \text{Top-}k(\{r(v_i, v_k)\}_{k=1}^{|\mathcal{V}|}), \\ 0, & \text{otherwise}, \end{cases} \tag{6}$$

where Top-$k(\cdot)$ selects the $k$ largest values in $\{r(v_i, v_j)\}_{j=1}^{|\mathcal{V}|}$ according to the relevance scores. This ensures that only the most relevant connections are retained in the constructed labeled-unlabeled graph $G$. In concrete, $G$ enables us to capture the relationships among both labeled and unlabeled samples, which is critical for selecting impactful unlabeled samples for pseudo-labeling.

## 3.4. Selecting Unlabeled Samples for Pseudo-Labeling

Notably, due to the prohibitive cost of labeling, the number of labeled samples is significantly smaller than that of unlabeled samples, i.e., $|\mathcal{D}_L| \ll |\mathcal{D}_U|$. To complement the limited labeled information in $\mathcal{D}_L$, we propose to leverage the concept of *node influence*, which describes the extent to which the representation of a node can be impacted by another node (Xu et al., 2018; Huang & Zitnik, 2020; Wang & Leskovec, 2020). We first provide the formal definition as follows:

**Definition 3.1. [Node Influence]** The node influence from node $v_i$ to node $v_j$ is defined as $I(v_i, v_j) = \|\partial \mathbf{v}_i / \partial \mathbf{v}_j\|$, where $\mathbf{v}_i$ and $\mathbf{v}_j$ are the node representations of $v_i$ and $v_j$ learned by the widely adopted neighborhood aggregation mechanism, respectively. $\partial \mathbf{v}_i / \partial \mathbf{v}_j$ is a Jacobian matrix, and the norm can be any specific subordinate norm.

According to Definition 3.1, larger node influence indicates that the representation of a node can more easily affect another node, i.e., having a higher influence. We propose to consider unlabeled samples that have a higher influence on the entire set of labeled samples because these high-

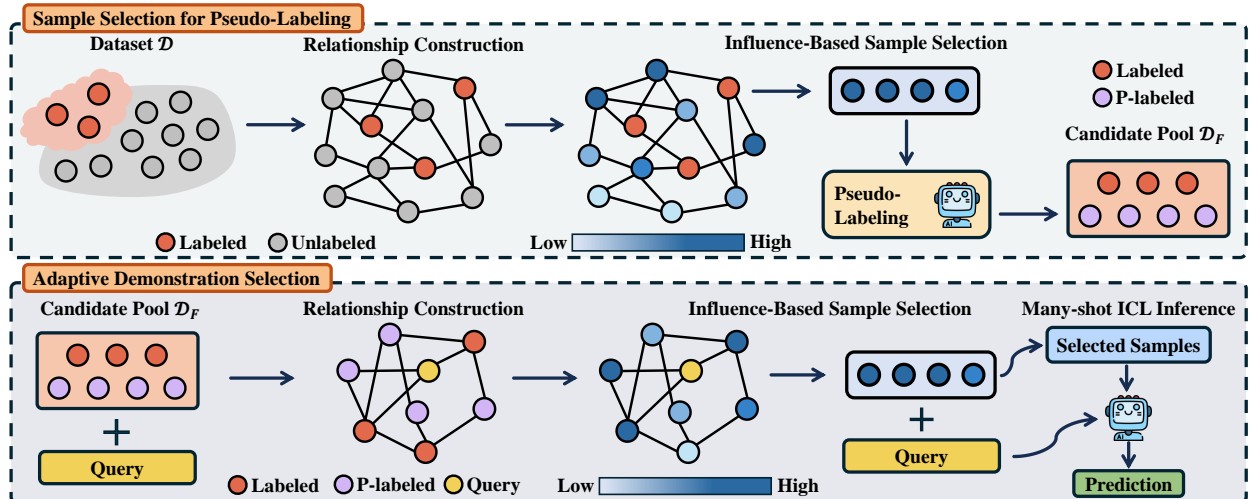

Figure 2: Overview of the MAPLE framework. Given a dataset with a small fraction of labeled samples, we select unlabeled samples for pseudo-labeling using the proposed influence score. During inference, we adopt a similar approach to select relevant samples from the candidate pool for each query, filtering out unrelated ones. The remaining samples are then used for many-shot ICL.

influence unlabeled samples are inherently more representative of the underlying data distribution. By pseudo-labeling these impactful samples, we enrich the demonstration pool with samples that reflect critical patterns and relationships within the specific dataset. We denote the node set as $\mathcal{V} = \mathcal{V}_L \cup \mathcal{V}_U$, where $\mathcal{V}_L$ and $\mathcal{V}_U$ correspond to samples in $\mathcal{D}_L$ and $\mathcal{D}_U$, respectively.

To effectively estimate the node influence of an unlabeled sample on the entire set of labeled samples, we propose the following theorem that provides a lower bound for the node influence on any set of nodes $\mathcal{V}$ in $G$:

**Theorem 3.2.** *Consider the node influence from node $u$ to a node set $\mathcal{V}$. Denote the geometric mean of the node influence to all nodes in $\mathcal{V}$ as $I_{\mathcal{V}}(u) = \sqrt[|\mathcal{V}|]{\prod_{i=1}^{|\mathcal{V}|} I(u, v_i)}$, where $v_i$ is the $i$-th node in $\mathcal{V}$. Assume the node degrees are randomly distributed with the mean value as $d$. Then,*

$$\mathbb{E}(\log I_{\mathcal{V}}(u)) \geq \log \widetilde{P_S}(u, \mathcal{V}) - \log d \cdot \overline{L}_S(u, \mathcal{V}), \quad (7)$$

*where $\overline{L}_S(u, \mathcal{V})$ is the average shortest path distance between $u$ and nodes in $\mathcal{V}$. $\widetilde{P_S}(u, \mathcal{V})$ is the geometric mean of the numbers of shortest paths between $u$ and nodes in $\mathcal{V}$.*

We provide the proof in Appendix A. From Theorem 3.2, we can conclude that in order to select nodes that are most influential to a set of nodes, we need to consider both the shortest path distance and the number of shortest paths.

Specifically, for each unlabeled sample $x \in \mathcal{D}_U$, which corresponds to $v \in \mathcal{V}_U$, we define its influence score on the labeled set $\mathcal{D}_L$ as

$$s(\mathcal{V}_L, v) = \log \widetilde{P_S}(v, \mathcal{V}_L) - \log d \cdot \overline{L}_S(v, \mathcal{V}_L), \quad (8)$$

where $\mathcal{V}_U$ and $\mathcal{V}_L$ denote the node sets of unlabeled samples and labeled samples, respectively.

We rank all unlabeled nodes and select the top $P$ nodes with the highest values of $s(\mathcal{V}_L, v)$ for pseudo-labeling, denoted as $\mathcal{V}_U^*$:

$$\mathcal{V}_U^* = \operatorname*{argmax}_{v \in \mathcal{V}_U, |\mathcal{V}_U^*|=P} \sum_{v \in \mathcal{V}_U} s(\mathcal{V}_L, v). \quad (9)$$

The pseudo-labels for selected samples in $\mathcal{V}_U^*$ are generated using a language model $\mathcal{M}_p$:

$$\hat{y}_j = \mathcal{M}_p(x_j), \quad \forall x_j \in \mathcal{D}_U^*, \quad (10)$$

where $\mathcal{D}_U^*$ denotes the samples that corresponding to nodes in $\mathcal{V}_U^*$. The pseudo-labeled samples, along with the assigned pseudo-labels, are then aggregated with the labeled set $\mathcal{D}_L$ to constitute the final candidate demonstration pool $\mathcal{D}_F$:

$$\mathcal{D}_F = \{(x_j, \hat{y}_j) \mid v_j \in \mathcal{V}_U^*\} \cup \mathcal{D}_L. \quad (11)$$

This process ensures that the most influential unlabeled samples are identified and pseudo-labeled, resulting in a high-quality demonstration pool from which the demonstrations for all input queries can be selected.

### 3.5. Adaptive Demonstration Selection

Now $\mathcal{D}_F$ contains only labeled and pseudo-labeled samples, which exhibit high influence and can serve as demonstrations for many-shot ICL. However, it still remains unsolved which samples will ultimately contribute meaningfully to the prediction of a specific test query $x_{\text{test}}$ by the LLM. To adaptively select the final demonstration set from $\mathcal{D}_F$, we again utilize the concept of node influence. We first establish a pseudo-labeled graph $G'(x_{\text{test}}) = (\mathcal{V}', \mathcal{E}')$ tailored

for each test query $x_{\text{test}}$. $G'(x_{\text{test}})$ consists of all samples in $\mathcal{D}_F$, along with $x_{\text{test}}$, thereby $|\mathcal{V}'| = |\mathcal{D}_F| + 1$. The edges $\mathcal{E}'$ in $G'$ are constructed in the same pattern as the labeled-unlabeled graph, leveraging the relevance scores between any pair of nodes. Notably, as $G'$ involves labeled samples and pseudo-labeled samples, we calculate the relevance scores based on both the queries (i.e., $x$) and labels (i.e., $y$):

$$\tilde{r}(v_i, v_j) = \begin{cases} \langle f_\theta(x_i, \hat{y}_i), f_\theta(x_j, \hat{y}_j) \rangle, & \text{if } x_j \neq x_{\text{test}}, \\ \langle f_\theta(x_i), f_\theta(x_j) \rangle, & \text{otherwise,} \end{cases}$$
(12)

With the calculated relevance scores, we construct the pseudo-labeled graph $G'(x_{\text{test}})$. The adjacency matrix is obtained as follows:

$$\mathbf{A}'_{ij} = \begin{cases} \tilde{r}(v_i, v_j), & \text{if } \tilde{r}(v_i, v_j) \in \text{Top-}k\left(\{\tilde{r}(v_i, v_j)\}_{j=1}^{|\mathcal{V}'|}\right), \\ 0, & \text{otherwise,} \end{cases}$$
(13)

With the pseudo-labeled graph, we evaluate the influence of the labeled and pseudo-labeled samples on the test query $x_{\text{test}}$ to select demonstrations that are highly relevant to the query. We aim to select nodes in $\mathcal{V}'$ whose influence on the test node exceeds the average influence of labeled nodes. Formally, we estimate the influence of each node $v \in \mathcal{V}' \setminus \{v_{\text{test}}\}$ on $v_{\text{test}}$, based on Theorem 3.2. Notably, the calculation becomes a specific case in Theorem 3.2, where the node set $\mathcal{V}$ only involves one node. The score is denoted as $s(v, v_{\text{test}})$. In this manner, the demonstration set $\mathcal{S}(x_{\text{test}})$ is then constructed as follows:

$$\mathcal{S}(x_{\text{test}}) = \underset{\mathcal{V}_S \subseteq \mathcal{V}' \setminus \{v_{\text{test}}\}}{\operatorname{argmax}} \sum_{v \in \mathcal{V}_S} s(v, v_{\text{test}}),$$
(14)
$$\text{where } |\mathcal{V}_S| = \alpha \cdot |\mathcal{V}'|.$$

$\alpha \in \mathbb{R}$ is a hyperparameter to control the percentage of samples we would like to include in $\mathcal{S}(x_{\text{test}})$. The final demonstration set $\mathcal{S}(x_{\text{test}})$ is then used as input to the pre-trained language model $\mathcal{M}$ for predicting the output of $x_{\text{test}}$:

$$\hat{y}_{\text{test}} = \mathcal{M}(\mathcal{S}(x_{\text{test}}), x_{\text{test}}).$$
(15)

### 3.6. Computation Cost Analysis

Our proposed method, MAPLE, performs graph construction in both selecting relevant unlabeled samples and adaptive demonstration selection. In this section, we analyze the computational cost associated with graph construction. Given a graph $\mathcal{G} = (\mathcal{V}, \mathcal{E})$, the graph construction requires the computation of the relevance score $r$ among any pair of nodes, which will be $\mathcal{O}(|\mathcal{V}|^2)$. To compute shortest paths, we use breadth-first search for each node, and the cost is $\mathcal{O}(|\mathcal{V}| + |\mathcal{E}|) = \mathcal{O}(|\mathcal{V}|)$ as $\mathcal{E} = \mathcal{O}(k|\mathcal{V}|)$. Therefore, the whole shortest path computation cost is $\mathcal{O}(|\mathcal{D}_L||\mathcal{V}|)$. Notably, the above cost is only required *once* before inference,

and does not scale with the number of test-time queries. With more queries involved during the test, the computational cost of the graph becomes negligible. As for adaptive demonstration selection, we further note that adaptive demonstration selection is an optional component that offers a trade-off between efficiency and performance, which will be discussed in Section 4.5.

## 4. Experiments

### 4.1. Experimental Settings

**Baselines.** In our experiments, we consider baselines that focus on how to select unlabeled demonstrations for pseudo-labeling to further improve many-shot ICL performance, given a fixed set of labeled instances. We compare our approach against the following baseline methods: (1) **Zero-shot**: This method provides only the task instruction to the LLM, without any demonstrations. (2) **Few-shot**: This approach incorporates only the labeled demonstrations for the LLM. (3) **Random**: This method randomly selects demonstrations for pseudo-labeling. (4) **RAG**: This approach uses Contriever (Izacard et al., 2021) to compute embeddings of both the query and the unlabeled demonstrations, selecting the most similar demonstrations for pseudo-labeling. (5) **RAG-Adapt**: This approach adds an adaptive demonstration selection process based on RAG based on embeddings of the query and the candidate demonstration pool $\mathcal{D}_F$.

**Datasets.** We evaluate the effectiveness of our approach on eight datasets across four tasks. (1) **Summarization**: This task assesses the ability of models to generate concise and coherent summaries from articles. We use the widely adopted XSum dataset (Narayan et al., 2018) for evaluation, with the ROUGE-L score as our evaluation metric. (2) **Reasoning**: This task tests the models' capacity for complex reasoning. We evaluate performance on three challenging datasets (Date, Salient, and Tracking7) from the Big Bench Hard (BBH) (Suzgun et al., 2023), following the experimental setup of the Long-Context Frontiers (LOFT) benchmark (Lee et al., 2024). (3) **Classification**: In this task, we focus on Financial PhraseBank (FP) sentiment analysis (Malo et al., 2014; Wei & Liu, 2025) and a subset of challenging benchmark datasets (Li et al., 2024) that are specifically designed for ICL tasks with diverse classes and long inputs, including Banking77 and GoEmotion. (4) **Question Answering**: We evaluate performance on the Google-Proof QA (GPQA) dataset (Rein et al., 2023), a multiple-choice QA benchmark. The questions are designed to challenge graduate-level reasoning in subjects such as biology and chemistry. The detailed descriptions of the used datasets are provided in Appendix C.

**Implementation.** In our main experiment, we set $k = 20$ and $\alpha = 0.75$, and we sample 1,000 demonstrations for

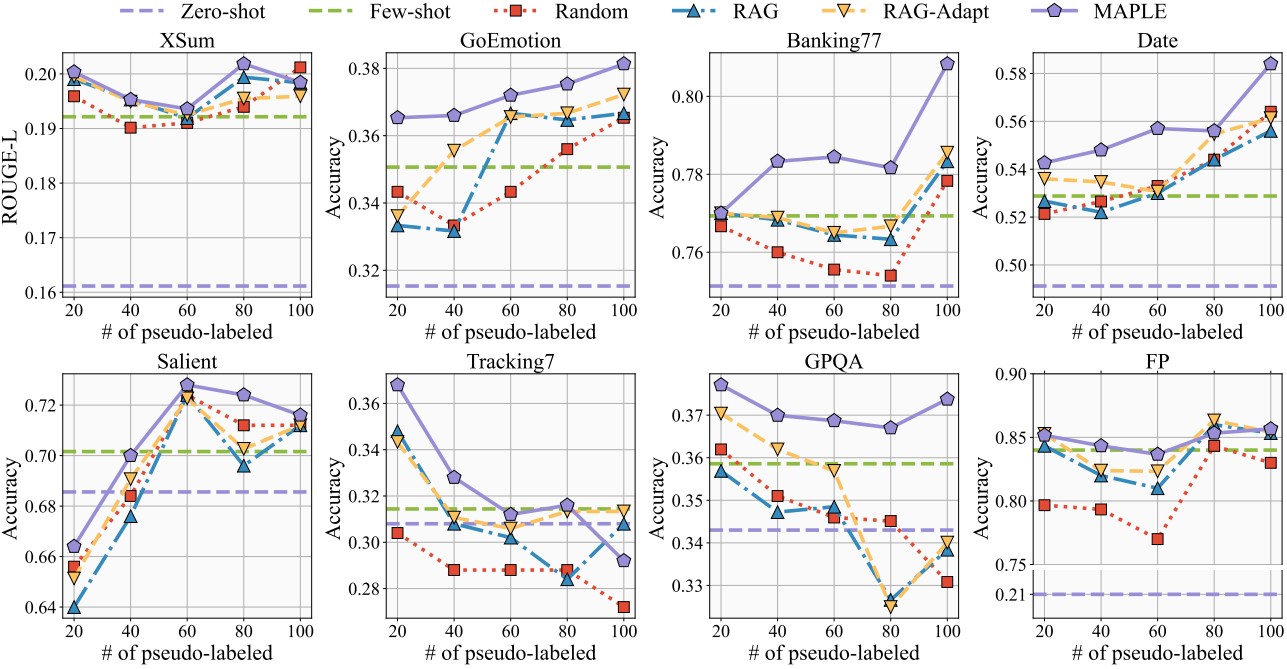

Figure 3: Performance comparison of various sample selection strategies in a many-shot ICL setting using Gemini 1.5 Flash across multiple datasets. 'Zero-shot' refers to the query LLM being provided only with the task instruction. We randomly selected 20 labeled samples to construct $\mathcal{D}_L$, and the results obtained using just $\mathcal{D}_L$ are presented as 'Few-shot'. Based on $\mathcal{D}_L$, we compare MAPLE with other pseudo-labeling baselines with different $\mathcal{D}_U^*$.

labeling and 300 for testing. For datasets with fewer than 1,000 training samples or fewer than 300 test samples, we use the entire dataset. We randomly select 20 demonstrations to form $\mathcal{D}_L$. Unless specified otherwise, we evaluate the many-shot ICL performance of the Gemini 1.5 Flash (Team et al., 2024) model with 1M token context length. We apply the Contriver (Izacard et al., 2021) as $f_\theta(\cdot)$. We conduct experiments with pseudo-labeled sizes ranging from 20 to 100. For most tasks, while increasing the number of demonstrations further improves performance, many-shot ICL reaches a sufficiently good performance with around 100 to $2^7$ demonstrations (Agarwal et al., 2024). The prompts used to elicit responses from ICL are provided in the Appendix B. Each experiment is run five times, and the average performance is reported.

### 4.2. Main Result

As shown in Fig. 3, we evaluate our method, MAPLE, in comparison to all baselines across eight datasets. From the results, we observe the following: ❶ **Superior Performance Across Datasets:** Our method consistently outperforms all other baselines across all eight datasets, showing significant improvements with various numbers of pseudo-labeled demonstrations. This highlights the effectiveness of our framework in selecting suitable unlabeled samples for pseudo-labeling in many-shot ICL. ❷ **Exceptional Results**

**on Complex Tasks:** MAPLE demonstrates particularly strong performance on complex datasets such as Banking77, Data, and GPQA. This suggests that in tasks requiring nuanced understanding, our framework's ability to select appropriate unlabeled samples provides a clear advantage over baseline methods. ❸ **Limited Gains in Certain Datasets:** The inclusion of additional pseudo-labeled samples does not always lead to performance improvements. For datasets like **Tracking7** and **XSum**, the limited benefits can be attributed to low-quality pseudo-labeled samples that cannot effectively assist in the inference process. ❹ **Effectiveness of ICL Settings:** Across all datasets, the few-shot setting consistently outperforms the zero-shot setting, demonstrating the value of in-context learning. Furthermore, involving more pseudo-labeled samples selected by our framework leads to notable performance enhancements, underscoring the effectiveness of leveraging pseudo-labeled data in many-shot ICL. ❺ **Performance Degradation with More Pseudo-labeled Demonstrations:** For certain tasks, such as **Tracking77** and **Salient**, we observe a performance drop as the number of pseudo-labeled demonstration increases. This can be attributed to the fact that as more demonstrations are included, the influence of noisy labels becomes more pronounced. Additionally, each query does not necessarily require a large number of samples, and the inclusion of irrelevant examples can degrade performance.

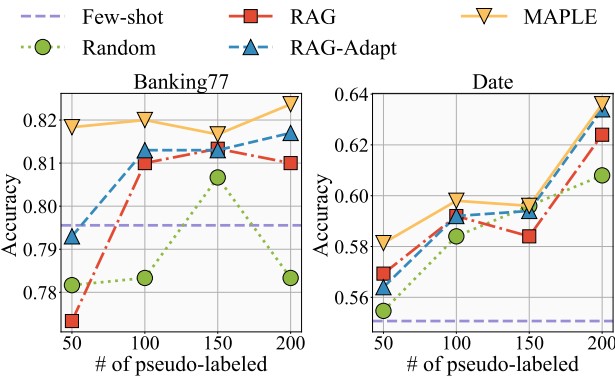

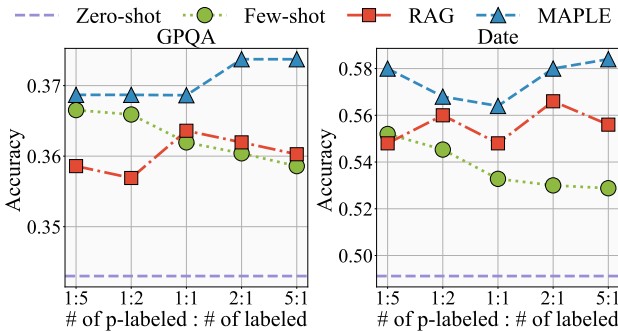

Figure 4: Results of MAPLE compared to baselines on two datasets with a larger number of demonstrations. We increase the size of $\mathcal{D}_L$ to 100 and compare performance for different sizes of $\mathcal{D}_U^*$ (50, 100, 150, and 200).

Figure 5: The results of varying the fraction of pseudo-labeled demonstrations. We fix the size of the candidate demonstration pool $\mathcal{D}_F$ as 120 and adjust the proportion of pseudo-labeled samples, while randomly selecting the labeled samples.

### 4.3. Impact of More Demonstrations

Given that long-context LLMs, such as Gemini 1.5 Flash, can process over 100 million tokens, we extend our experiments by increasing the number of labeled demonstrations to 100, while varying the number of pseudo-labeled demonstrations from 50 to 250. According to results presented in Fig. 4, we first observe that incorporating a larger number of demonstrations leads to improved performance, which validates the effectiveness of many-shot ICL. Furthermore, MAPLE consistently outperforms all baselines as the number of demonstrations increases, indicating the scalability of MAPLE in scenarios with a large number of demonstrations.

### 4.4. Results of Different LLMs

In our main experiments, we use Gemini 1.5 Flash. In this section, we adopt a stronger LLM, Gemini 1.5 Pro, to showcase the performance of MAPLE under different LLMs. The experimental results in Table 1 highlight several key observations. First, the proposed MAPLE method consistently outperforms the baseline across all tasks and LLM variants, demonstrating its ability to effectively leverage pseudo-labeled demonstrations. For instance, in the Banking77 task, MAPLE achieves an average accuracy improvement of 1.7% with Gemini Pro and 1.5% with Gemini Flash compared to the baseline. Second, MAPLE exhibits a notable advantage in scalability, showing steady improvements in performance as the number of pseudo-labeled demonstrations increases. This trend is particularly evident in the Date and GPQA tasks, where MAPLE achieves its highest accuracy at 100 demonstrations, outperforming the baseline by a significant margin. Finally, the stronger Gemini Pro model consistently amplifies the performance of MAPLE across all tasks, indicating that the method's benefits are further enhanced when paired with more advanced LLMs. These observations validate the robustness and adaptability of MAPLE in varying contexts.

### 4.5. KV Cache Ablation

In MAPLE, we provide personalized demonstrations for each query as the many-shot ICL prompt, which can be time-consuming during inference. In this section, we examine a variant of MAPLE that omits adaptive demonstration selection. In this setup, the labeled and pseudo-labeled demonstrations are fixed across all queries, allowing us to apply the KV Cache (Pope et al., 2023) to cache the demonstrations in the LLM before inference for improved efficiency. To illustrate the effect of KG Cache in our framework, we provide a detailed analysis regarding the FLOPs with KV Cache in Appendix D.

Furthermore, we investigate the trade-off between efficiency and performance of our framework MAPLE when using KV Cache. Particularly, we conduct experiments on datasets XSum and Date. We additionally consider using various numbers of pseudo-labeled demonstrations, and report the performance and inference time. From the results presented in Table 2, we observe that applying KV Cache can reduce inference time as expected. Moreover, the efficient improvement is more significant with a larger number of pseudo-labeled demonstrations. However, including more demonstrations as input for each query introduces more noise and irrelevant information, which negatively impacts performance. Moreover, we also note that the efficiency gains from KV Cache are not significant. This is potentially due to the time required for the internal loading of the KV cache in API-based models, such as Gemini 1.5 Flash.

### 4.6. Fraction of Pseudo-labeled Demonstrations

In this subsection, we conduct experiments to evaluate the impact of noisy pseudo-labeled data on many-shot ICL performance. In this manner, we further assess the quality of demonstrations selected by MAPLE in comparison to other baselines. In particular, we fix the total number of

Table 1: Results of many-shot ICL across three tasks (extreme classification, reasoning, and question answering) with different LLMs (Gemini 1.5 Pro and Gemini 1.5 Flash). We report the performance in % under different numbers of $|\mathcal{D}_U^*|$. MAPLE consistently outperforms the baseline across all tasks and LLM variants, demonstrating its ability to effectively leverage pseudo-labeled demonstrations.

| Task | Model | Method | # of Pseudo-labeled Demonstrations | | | | | Avg. |
|------|-------|--------|------|------|------|------|------|------|
| | | | 20 | 40 | 60 | 80 | 100 | |
| Banking77 | Gemini Flash | Random | $76.7 \pm 3.9$ | $76.0 \pm 4.4$ | $75.6 \pm 3.1$ | $75.4 \pm 3.7$ | $77.8 \pm 3.2$ | 76.3 |
| | | RAG | $77.0 \pm 2.3$ | $76.8 \pm 2.3$ | $76.4 \pm 2.4$ | $76.3 \pm 3.4$ | $78.3 \pm 4.1$ | 77.0 |
| | | MAPLE | $77.0 \pm 2.8$ | $78.3 \pm 2.6$ | $78.4 \pm 2.6$ | $78.1 \pm 2.7$ | $80.8 \pm 3.4$ | **78.5** |
| | Gemini Pro | Random | $74.7 \pm 2.3$ | $76.7 \pm 1.9$ | $79.3 \pm 2.2$ | $78.5 \pm 2.0$ | $78.3 \pm 2.1$ | 77.5 |
| | | RAG | $76.3 \pm 2.5$ | $78.3 \pm 2.0$ | $78.7 \pm 1.5$ | $79.3 \pm 2.8$ | $78.8 \pm 2.5$ | 78.3 |
| | | MAPLE | $79.3 \pm 2.2$ | $79.7 \pm 1.5$ | $80.3 \pm 2.3$ | $81.3 \pm 1.5$ | $80.7 \pm 1.9$ | **80.3** |
| Date | Gemini Flash | Random | $52.1 \pm 1.4$ | $52.7 \pm 1.5$ | $53.3 \pm 0.6$ | $54.4 \pm 1.6$ | $56.4 \pm 1.0$ | 53.8 |
| | | RAG | $52.7 \pm 1.0$ | $52.2 \pm 1.5$ | $53.0 \pm 0.9$ | $54.4 \pm 1.4$ | $55.6 \pm 1.8$ | 53.6 |
| | | MAPLE | $54.3 \pm 1.2$ | $54.8 \pm 1.3$ | $55.7 \pm 0.9$ | $55.6 \pm 1.8$ | $58.4 \pm 2.4$ | **55.8** |
| | Gemini Pro | Random | $64.4 \pm 1.0$ | $66.2 \pm 1.2$ | $66.8 \pm 1.8$ | $67.2 \pm 1.1$ | $68.0 \pm 0.8$ | 66.5 |
| | | RAG | $66.4 \pm 1.6$ | $66.8 \pm 1.5$ | $66.0 \pm 1.2$ | $67.2 \pm 0.5$ | $68.4 \pm 1.4$ | 67.0 |
| | | MAPLE | $67.6 \pm 0.8$ | $67.6 \pm 1.7$ | $67.2 \pm 0.6$ | $68.8 \pm 1.0$ | $68.8 \pm 3.0$ | **68.0** |
| GPQA | Gemini Flash | Random | $36.2 \pm 0.6$ | $35.1 \pm 1.9$ | $34.6 \pm 3.1$ | $34.5 \pm 2.3$ | $33.1 \pm 2.0$ | 34.7 |
| | | RAG | $35.7 \pm 1.2$ | $34.7 \pm 1.3$ | $34.8 \pm 0.5$ | $32.7 \pm 1.4$ | $33.8 \pm 1.0$ | 34.3 |
| | | MAPLE | $37.7 \pm 1.7$ | $37.0 \pm 0.4$ | $36.9 \pm 1.3$ | $36.7 \pm 1.9$ | $37.4 \pm 1.6$ | **37.1** |
| | Gemini Pro | Random | $41.4 \pm 0.8$ | $41.4 \pm 1.3$ | $43.3 \pm 2.0$ | $42.4 \pm 2.0$ | $42.9 \pm 1.1$ | 42.3 |
| | | RAG | $43.9 \pm 0.3$ | $43.3 \pm 2.5$ | $41.9 \pm 1.5$ | $41.9 \pm 1.6$ | $43.3 \pm 1.0$ | 42.9 |
| | | MAPLE | $44.4 \pm 2.3$ | $44.4 \pm 1.0$ | $44.9 \pm 2.0$ | $43.8 \pm 0.6$ | $43.9 \pm 1.6$ | **44.3** |

Table 2: Comparison of MAPLE with its variant, which removes the adaptive demonstration selection component. We report the average performance and the inference time (s) per query on two datasets XSum and Date.

| Task | KV | # of Pseudo-labeled | | |
|------|-----|------|------|------|
| | | 20 | 60 | 100 |
| XSum | ✓ | 0.200/1.89 | 0.191/3.57 | 0.198/4.08 |
| | - | 0.199/1.93 | 0.189/3.74 | 0.195/4.50 |
| | | 100 | 250 | 300 |
| Date | ✓ | 0.584/0.94 | 0.596/1.92 | 0.636/1.96 |
| | - | 0.578/0.91 | 0.589/1.99 | 0.622/2.04 |

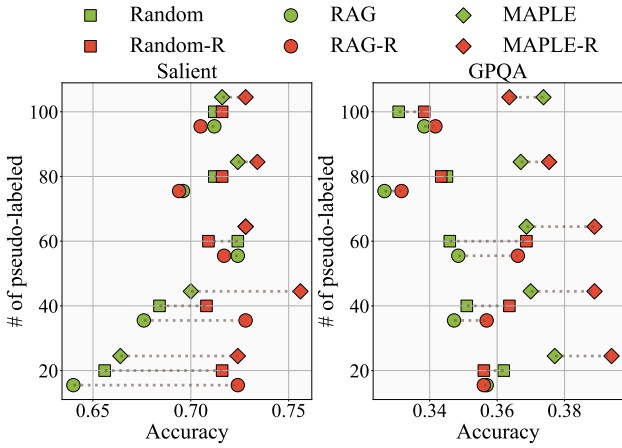

Figure 6: Impact of the order of the two sources of samples. Green dots represent methods with the default order, where labeled demonstrations appear first, and pseudo-labeled demonstrations are positioned closer to the query. Red dots represent methods with the order reversed, where pseudo-labeled demonstrations appear first and labeled demonstrations are positioned at the back.

demonstrations at 120 and adjust the fraction of labeled demonstrations versus pseudo-labeled demonstrations (ranging from 20 to 100 labeled). The results are presented in Fig. 5. We can obtain the following observations: (1) The experimental results demonstrate that MAPLE consistently outperforms baseline methods across all fractions of labeled and pseudo-labeled demonstrations, highlighting its effectiveness in selecting high-quality pseudo-labeled data. (2) Increasing the fraction of labeled demonstrations generally improves performance, as higher-quality labeled data provides stronger guidance. (3) When the number of labeled demonstrations decreases, appropriately selected pseudo-labeled data compensates for the loss, maintaining or even improving performance. This is particularly evident in cases where pseudo-labeled data provides additional information

to offset the reduction in labeled examples, showcasing the robustness and adaptability of MAPLE in leveraging both labeled and pseudo-labeled demonstrations effectively.

## 4.7. Study of the Order of Demonstrations

In our main experiments, we put labeled demos in the front of the input and pseudo-labeled demos in the back of the input and closer to the query. We examine how the or-

der of labeled and pseudo-labeled demonstrations impacts many-shot ICL performance. The baseline setup places labeled demonstrations at the front and pseudo-labeled ones closer to the query, while the alternative swaps their positions. Results in Fig. 6 reveal that placing labeled demonstrations near the query generally improves performance, highlighting the importance of proximity to high-quality data. However, this improvement diminishes as the number of pseudo-labeled demonstrations increases, indicating that the influence of noisy pseudo-labeled data becomes more dominant with higher proportions. These findings emphasize the significance of demonstration order in enhancing ICL outcomes.

## 5. Conclusion

In this work, we focus on enhancing many-shot ICL performance in resource-constrained tasks. we propose a novel adaptive pseudo-labeling framework for many-shot ICL, which selects the most impactful unlabeled samples for pseudo-labeling. Additionally, we introduce an adaptive method to select both pseudo-labeled and labeled samples as demonstrations for LLM input. Extensive experiments across various datasets validate the effectiveness of our framework. In future work, we aim to further explore the interaction between labeled and pseudo-labeled samples and its impact on ICL performance. We believe this will help identify the most useful samples for pseudo-labeling.

## Acknowledgements

This work is supported in part by the National Science Foundation (NSF) under grants IIS-2006844, IIS-2144209, IIS-2223769, CNS-2154962, BCS-2228534, CMMI-2411248, ECCS-2029978, CPS-2313110, ECCS-2143559, AST-2132700, and ECCS-2033671; the Office of Naval Research (ONR) under grant N000142412636; and the Commonwealth Cyber Initiative (CCI) under grant VV-1Q24-011, and the research gift funding from Kneron.

## Impact Statement

This paper presents work whose goal is to improve the performance of many-shot ICL in resource-constrained settings where manual labeling is expensive or infeasible. There are many potential societal consequences of our work, none of which we feel must be specifically highlighted here.

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

# A. Theorem 3.2 and Proof

Prior to proving Theorem 3.2, we present a lemma that establishes the lower bound of node influence between two nodes. In the subsequent proof, we adopt the approach outlined in (Huang & Zitnik, 2020) and (Xu et al., 2018), utilizing GCNs (Kipf & Welling, 2017) as the exemplar GNN for simplicity. It is worth noting that our proof can be readily extended to different types of models (such as GAT (Veličković et al., 2018) and GraphSAGE (Hamilton et al., 2017)) by assigning different values to edge weights. Specifically, the propagation process in the $l$-th layer can be represented as $\mathbf{H}^{(l+1)} = \sigma(\hat{\mathbf{A}}\mathbf{H}^{(l)}\mathbf{W}^{(l)})$, where $\mathbf{H}^{(l)}$ and $\mathbf{W}^{(l)}$ denote the node representation and weight parameter matrices, respectively. $\hat{\mathbf{A}} = \mathbf{D}^{-1}\mathbf{A}$ represents the adjacency matrix after row normalization, ensuring that each row of $\hat{\mathbf{A}}$ sums up to 1. Following the convention of Huang & Zitnik (2020), Wang & Leskovec (2020), and Xu et al. (2018), we set $\sigma$ as the identity function and $\mathbf{W}$ as the identity matrix. Additionally, we assume that the propagation process is performed over a sufficient number of iterations. Consequently, the output representation of a node can be expressed as a function of the representations of its neighboring nodes.

**Lemma A.1.** *Consider the log-expectation of node influence between node $v_i$ and node $v_j$, i.e., $\mathbb{E}\left(\log\left(I(v_i, v_j)\right)\right)$. Assume that the node degrees are distributed uniformly for each node with the mean value d. Then, $\mathbb{E}\left(\log\left(I(v_i, v_j)\right)\right) \geq \log P_S(v_i, v_j) - L_S(v_i, v_j) \cdot \log d$, where $L_S(v_i, v_j)$ is the shortest path distance between $v_i$ and $v_j$, and $P_S(v_i, v_j)$ is the number of paths with length of $L_S(v_i, v_j)$ from $v_i$ to $v_j$.*

*Proof.* According to the propagation strategy used in GCNs, we acknowledge that the representation of node $v_i$ can be represented as

$$\mathbf{h}_i = \frac{1}{D_{ii}} \sum_{k \in \mathcal{N}(i)} a_{ik} \mathbf{h}_k,$$

where $\mathcal{N}(i)$ denotes the set of neighboring nodes of node $v_i$. After that, we can expand the equation via incorporating more neighbors of node $v_i$:

$$
\begin{aligned}
\mathbf{h}_i &= \frac{1}{D_{ii}} \sum_{k \in \mathcal{N}(i)} a_{ik} \frac{1}{D_{kk}} \sum_{l \in \mathcal{N}(k)} a_{kl} \mathbf{h}_l \\
&= \frac{1}{D_{ii}} \sum_{k \in \mathcal{N}(i)} a_{ik} \frac{1}{D_{kk}} \sum_{l \in \mathcal{N}(k)} a_{kl} \cdots \frac{1}{D_{mm}} \sum_{o \in \mathcal{N}(m)} a_{mo} \mathbf{h}_o.
\end{aligned}
\tag{16}
$$

In this manner, the node influence $I_{i,j} = \|\partial \mathbf{h}_i / \partial \mathbf{h}_j\|$ can be represented as:

$$
\begin{aligned}
\left\| \frac{\partial \mathbf{h}_i}{\partial \mathbf{h}_j} \right\| &= \left\| \frac{\partial}{\partial \mathbf{h}_j} \left( \frac{1}{D_{ii}} \sum_{k \in \mathcal{N}(i)} a_{ik} \frac{1}{D_{kk}} \sum_{l \in \mathcal{N}(k)} a_{kl} \cdots \frac{1}{D_{mm}} \sum_{o \in \mathcal{N}(m)} a_{mo} \mathbf{h}_o \right) \right\| \\
&= \left\| \frac{\partial}{\partial \mathbf{h}_j} \left( \left( \frac{1}{D_{ii}} a_{ik_1^1} \frac{1}{D_{k_1^1 k_1^1}} a_{k_1^1 k_2^1} \cdots \frac{1}{D_{k_{n_1}^1 k_{n_1}^1}} a_{k_{n_1}^1 j} \mathbf{h}_j \right) \right. \right. \\
&\qquad \left. \left. + \cdots + \left( \frac{1}{D_{ii}} a_{ik_1^n} \frac{1}{D_{k_1^n k_1^n}} a_{k_1^n k_2^n} \cdots \frac{1}{D_{k_{n_n}^n k_{n_n}^n}} a_{k_{n_n}^n j} \mathbf{h}_j \right) \right) \right\|.
\end{aligned}
\tag{17}
$$

In the above derivation, we begin by replacing the term $\mathbf{h}_i$ with the iterative expansion of its neighboring nodes. This expansion involves selecting only $n$ paths from $v_i$ and to $v_j$, where $n_i$ represents the number of intermediate nodes along the $i$-th path. This selection is motivated by the fact that, when considering the gradient between $v_i$ and $v_j$, the derivatives along paths that do not include $v_j$ would be 0 and thus can be disregarded. Subsequently, we proceed to extract the shared

term $\|\partial \mathbf{h}_j / \partial \mathbf{h}_j\|$:

$$
\begin{aligned}
\left\|\frac{\partial \mathbf{h}_i}{\partial \mathbf{h}_j}\right\| &= \left\|\frac{\partial \mathbf{h}_j}{\partial \mathbf{h}_j}\right\| \cdot \left( \left( \frac{1}{D_{ii}} a_{ik_1^1} \frac{1}{D_{k_1^1 k_1^1}} a_{k_1^1 k_2^1} \cdots \frac{1}{D_{k_{n_1}^1 k_{n_1}^1}} a_{k_{n_1}^1 j} \right) \right. \\
&\quad + \cdots + \left. \left( \frac{1}{D_{ii}} a_{ik_1^n} \frac{1}{D_{k_1^n k_1^n}} a_{k_1^n k_2^n} \cdots \frac{1}{D_{k_{n_n}^n k_{n_n}^n}} a_{k_{n_n}^n j} \right) \right) \\
&= \left( \frac{1}{D_{ii}} a_{ik_1^1} \frac{1}{D_{k_1^1 k_1^1}} a_{k_1^1 k_2^1} \cdots \frac{1}{D_{k_{n_1}^1 k_{n_1}^1}} a_{k_{n_1}^1 j} \right) \\
&\quad + \cdots + \left( \frac{1}{D_{ii}} a_{ik_1^n} \frac{1}{D_{k_1^n k_1^n}} a_{k_1^n k_2^n} \cdots \frac{1}{D_{k_{n_n}^n k_{n_n}^n}} a_{k_{n_n}^n j} \right).
\end{aligned}
\tag{18}
$$

In this derivation, we first employ the identity $\|\partial \mathbf{h}_j / \partial \mathbf{h}_j\| = 1$. This identity holds because $\|\partial \mathbf{h}_j / \partial \mathbf{h}_j\| = \|\mathbf{I}\| = \sup_{\|\mathbf{h}\|=1} \|\mathbf{Ih}\| = 1$. The resulting term represents an expectation that involves summing the products of node degrees along all paths connecting $v_i$ and $v_j$. As a result, it surpasses the values obtained by summing the products of node degrees along (potentially multiple) paths with the minimum node degree product:

$$
\begin{aligned}
\left\|\frac{\partial \mathbf{h}_i}{\partial \mathbf{h}_j}\right\| &\geq \left( \frac{1}{D_{ii}} a_{ik_1^{P_1}} \frac{1}{D_{k_1^{P_1} k_1^{P_1}}} a_{k_1^{P_1} k_2^{P_1}} \cdots \frac{1}{D_{k_{n_1}^{P_1} k_{n_1}^{P_1}}} a_{k_{n_1}^{P_1} j} \right) \\
&\quad + \cdots + \left( \frac{1}{D_{ii}} a_{ik_1^{P_m}} \frac{1}{D_{k_1^{P_m} k_1^{P_m}}} a_{k_1^{P_m} k_2^{P_m}} \cdots \frac{1}{D_{k_{n_{P_m}}^{P_m} k_{n_{P_m}}^{P_m}}} a_{k_{n_{P_m}}^{P_m} j} \right) \\
&= P_m \cdot \max \left( \left( \frac{1}{D_{ii}} a_{ik_1^1} \frac{1}{D_{k_1^1 k_1^1}} a_{k_1^1 k_2^1} \cdots \frac{1}{D_{k_{n_1}^1 k_{n_1}^1}} a_{k_{n_1}^1 j} \right) \right. \\
&\quad \left. , \cdots, \left( \frac{1}{D_{ii}} a_{ik_1^m} \frac{1}{D_{k_1^m k_1^m}} a_{k_1^m k_2^m} \cdots \frac{1}{D_{k_{n_m}^m k_{n_m}^m}} a_{k_{n_m}^m j} \right) \right).
\end{aligned}
\tag{19}
$$

Assuming that the node degrees are uniformly distributed, the expectation of node degree products on path $P_i$ is $d^{(n_{P_i}+1)}$, where $n_{P_i} + 1$ is the length, and $d$ is the expectation of node degrees. Furthermore, it is noteworthy that these paths are exactly the shortest paths between $v_i$ and $v_j$. Therefore,

$$
\mathbb{E}\left( \left\|\frac{\partial \mathbf{h}_i}{\partial \mathbf{h}_j}\right\| \right) \geq P_m \left( 1/d \right)^{(n_*+1)} = P_m d^{-(L_S(v_i, v_j))},
\tag{20}
$$

where $L_S(v_i, v_j)$ denotes the shortest path distance between node $v_i$ and node $v_j$, and $P_m$ is the number of these paths. Then we can achieve the final result:

$$
\mathbb{E}\left( \log \left( I(v_i, v_j) \right) \right) = \log \mathbb{E}\left( \left\|\frac{\partial \mathbf{h}_i}{\partial \mathbf{h}_j}\right\| \right) \geq \log P_S(v_i, v_j) - L_S(v_i, v_j) \cdot \log d,
\tag{21}
$$

where $P_S(v_i, v_j)$ is the number of paths with length of $L_S(v_i, v_j)$ from $v_i$ to $v_j$. $\qquad \square$

Lemma A.1 demonstrates that the expectation of logarithmic node influence between two nodes is related to two perspectives: the shortest path distance and the number of shortest path distances between them. Now with Lemma A.1, we can prove Theorem 3.2.

**Theorem 3.2.** *Consider the node influence from node $u$ to a node set $\mathcal{V}$. Denote the geometric mean of the node influence to all nodes in $\mathcal{V}$ as $I_{\mathcal{V}}(u) = \sqrt[|\mathcal{V}|]{\prod_{i=1}^{|\mathcal{V}|} I(u, v_i)}$, where $v_i$ is the $i$-th node in $\mathcal{V}$. Assume the node degrees are randomly distributed with the mean value as $d$. Then,*

$$
\mathbb{E}(\log I_{\mathcal{V}}(u)) \geq \log \widetilde{P_S}(u, \mathcal{V}) - \log d \cdot \overline{L}_S(u, \mathcal{V}),
\tag{22}
$$

*where $\overline{L}_S(u, \mathcal{V})$ is the average shortest path distance between $u$ and nodes in $\mathcal{V}$. $\widetilde{P_S}(u, \mathcal{V})$ is the geometric mean of the numbers of shortest paths between $u$ and nodes in $\mathcal{V}$.*

*Proof.* We know $\log I_{\mathcal{V}}(v_k)$ can be represented as follows:

$$\log I_{\mathcal{V}}(u) = \frac{1}{|\mathcal{V}|} \sum_{i=1}^{|\mathcal{V}|} \log I(u, v_i). \tag{23}$$

Based on Lemma A.1, we know:

$$
\begin{aligned}
\mathbb{E}\left(\log I_{\mathcal{V}}(v_k)\right) &= \frac{1}{|\mathcal{V}|} \sum_{i=1}^{|\mathcal{V}|} \log \mathbb{E}\left(I(u, v_i)\right) \\
&\geq \frac{1}{|\mathcal{V}|} \cdot \sum_{i=1}^{|\mathcal{V}'|} \left(\log P_S(u, v_i) - L_S(u, v_i) \cdot \log d\right),
\end{aligned}
\tag{24}
$$

where $L_S(u, v_i)$ denotes the shortest path distance between node $u$ and node $v_i$, and $P_S(u, v_i)$ is the number of the shortest paths between $u$ and $v_i$. Note that $P_S(u, v_i) \geq 1$, as any connected node pair should have at least a path. By rearranging the term, we can obtain the final inequality:

$$
\begin{aligned}
\mathbb{E}(\log I_{\mathcal{V}}(u)) &\geq \frac{1}{|\mathcal{V}|} \sum_{i=1}^{|\mathcal{V}|} \log P_S(u, v_i) - \log d \cdot \frac{1}{|\mathcal{V}|} \sum_{i=1}^{|\mathcal{V}|} L_S(u, v_i) \\
&= \log \left(\prod_{i=1}^{|\mathcal{V}|} P_S(u, \mathcal{V})\right)^{\frac{1}{n}} - \log d \cdot \overline{L}_S(u, \mathcal{V}) \\
&= \log \widetilde{P_S}(v_i, \mathcal{V}) - \log d \cdot \overline{L}_S(v_i, \mathcal{V}).
\end{aligned}
\tag{25}
$$

$\square$

# B. Prompts

We provide the prompts used for many-shot ICL in the summarization, reasoning, and question answering tasks in Table 3, and for classification tasks in Table 4.

Table 3: A list of prompts that we use for many-shot ICL on summarization, reasoning, and question answering tasks.

| Types | Prompts |
|---|---|
| Summarization | You are an expert in article summarization. I am going to give you some examples of article and its summary in fluent English. Here are several examples. 

 (provide examples here with the following format.) 
 Article: \<article\> 
 Summary: \<summary\> 

 I am going to provide another article and I want you to summarize it. Give only the summary, and no extra commentary, formatting, or chattiness. 

 Article: {TARGET_QUERY} |
| Reasoning 

 or 

 Quesition Answering | You are an expert in multiple-choice question answering tasks. I am going to give you some examples in a multiple-choice question answering format. Here are several examples. 

 (provide examples here with the following format.) 
 Question: \<question\> 
 Answer: \<answer\> 

 I am going to provide another question and I want you to predict its answer. Give only the choice the correct answer by selecting one of the options (e.g., '(A)', '(B)'). 

 Question: {TARGET_QUERY} |

Table 4: A list of prompts that we use for many-shot ICL on five different extreme classification tasks.

| Types | Prompts |
|---|---|
| Financial PhraseBank | You are an expert in financial sentiment analysis. Here are several examples. 

 (provide examples here with the following format.) 
 Sentence: \<sentence\> 
 Answer: \<sentiment\> 

 I am going to provide another sentence and I want you to analyze the sentiment of it and respond with only one word: 'positive', 'negative', or 'neutral'. No extra commentary, formatting, or chattiness. 

 Sentence: {TARGET_QUERY} |
| Banking77 | Given a customer service query, please predict the intent of the query. Here are several examples. 

 (provide examples here with the following format.) 
 service query: \<query\> 
 intent category: \<category\> 

 I am going to provide another customer service query and I want you to predict the intent of the query. Give only the intent of the query, and no extra commentary, formatting, or chattiness. You can only make prediction from the following categories: {77 classes of the Banking77 task} 

 service query: {TARGET_QUERY} |
| GoEmotion | Given a comment, please predict the emotion category of this comment. Here are several examples. 

 (provide examples here with the following format.) 
 comment: \<comment\> 
 emotion category: \<category\> 

 I am going to provide another comment and I want you to predict the emotion category of the comment. Give only the emotion category, and no extra commentary, formatting, or chattiness. You can only make prediction from the following categories: {28 classes of the GoEmotion task} 

 comment: {TARGET_QUERY} |

## C. Datasets

- **XSum (Narayan et al., 2018):** Extreme Summarization (XSum) is a dataset designed for evaluating abstractive single-document summarization systems. It contains 226,711 news articles from BBC (2010-2017), spanning various domains such as news, politics, sports, weather, business, technology, science, health, family, education, and entertainment. The goal is to generate a concise one-sentence summary answering the question, "What is the article about?"

- **Date (Suzgun et al., 2023):** This dataset is designed for Date Understanding, where the task is to answer a provided question based on a small set of sentences related to a particular date. An example is: "Today is Christmas Eve of 1937. What is the date tomorrow in MM/DD/YYYY? Options: (A) 12/11/1937; (B) 12/25/1937; (C) 01/04/1938; (D) 12/04/1937; (E) 12/25/2006; (F) 07/25/1937"

- **Salient (Suzgun et al., 2023):** This dataset is designed for Salient Translation Error Detection, where, given a source sentence written in German and its English translation, the task is to determine the type of translation error present in the translated sentence. An example is: "Source: Karl Borrom Ŏ0e4us Joseph FŎ0fcrst von Liechtenstein war ein kaiserlicher Feldmarschall. Translation: Charles Borromeo Joseph Prince of Liechtenstein was an judicial field marshal. The translation contains an error pertaining to Options: (A) Modifiers or Adjectives; (B) Numerical Values; (C) Negation or Antonyms; (D) Named Entities; (E) Dropped Content; (F) Facts"

- **Tracking7 (Suzgun et al., 2023):** This dataset is designed for Tracking Shuffled Objects, where, given the initial positions of a set of seven objects and a series of transformations (specifically, pairwise swaps) applied to them, the goal is to determine the final positions of the objects. An example is: "Alice, Bob, Claire, Dave, Eve, Fred, and Gertrude are on the same team in a soccer match. At the start of the match, they are each assigned to a position: Alice is playing striker, Bob is playing right winger, Claire is playing left winger, Dave is playing benchwarmer, Eve is playing goalkeeper, Fred is playing center midfielder, and Gertrude is playing cheerleader. As the game progresses, pairs of players occasionally swap positions. First, Eve and Claire trade positions. Then, Gertrude and Alice trade positions. Then, Fred and Bob trade positions. Then, Dave and Fred trade positions. Then, Fred and Bob trade positions. Then, Bob and Eve trade positions. Finally, Claire and Alice trade positions. At the end of the match, Gertrude is playing: (A) striker; (B) right winger; (C) left winger; (D) benchwarmer; (E) goalkeeper; (F) center midfielder; (G) cheerleader"

- **Financial PhraseBank (FP) (Malo et al., 2014):** FP is a sentiment analysis dataset consisting of 4,840 sentences from English-language financial news, categorized by sentiment. The annotators were instructed to assess the sentences from an investor's perspective, determining whether the news would likely have a positive, negative, or neutral impact on stock prices. An example is: "Sentence: Pharmaceuticals group Orion Corp reported a fall in its third-quarter earnings, which were impacted by larger expenditures on R&D and marketing. Label: negative."

- **Banking77 (Casanueva et al., 2020):** BANKING77 is a dataset for banking-domain intent detection, comprising 13,083 annotated examples across 77 distinct intents. It offers a complex and realistic representation of commercial systems. An example is: "Text: I found my lost card. Am I still able to use it? Label: Link to Existing Card."

- **GoEmotion (Demszky et al., 2020):** GoEmotion is the largest human-annotated dataset, consisting of 58k carefully selected Reddit comments, labeled with 27 emotion categories or "Neutral." The comments are extracted from popular English subreddits. An example is: "Text: I'm not even sure what it is, why do people hate it. Label: confusion."

- **GPQA (Rein et al., 2023):** GPQA is a multiple-choice question answering benchmark that features challenging graduate-level questions in biology, physics, and chemistry, designed to assess advanced reasoning skills. An example question is: "If a sperm from species A is injected into an egg from species B, and both species have the same number of chromosomes, what would be the main cause of the resulting zygote's mortality?"

## D. KV Cache Analysis

We begin by providing a simple analysis of the FLOPs between the vanilla transformer and the transformer with KV Cache. We denote the length of the prefix context, model hidden size, the number of model layers, and the number of new tokens to generate as $n, d_{\text{model}}, L$, and $T$, respectively.

Table 5: Basic information and statistics of datasets adopted in our experiments.

| Dataset | Task | #(Training samples) | #(Test samples) |
|---------|------|---------------------|-----------------|
| XSum | Summarization | 204,045 | 11,334 |
| Date | Reasoning | 369 | 250 |
| Salient | Reasoning | 998 | 250 |
| Tracking7 | Reasoning | 1,750 | 250 |
| Financial PhraseBank | Classification | 2,952 | 500 |
| Banking77 | Classification | 10,003 | 3,080 |
| GoEmotion | Classification | 43,410 | 5,427 |
| GPQA | Question Answering | 250 | 198 |

**Vanilla decoding without KV Cache.** The per-step cost for a length-$m$ forward pass in one layer can be approximated as:

$$O\big(m\,d_{\text{model}}^2 + m^2\,d_{\text{model}}\big).$$

At each step $t$, we re-run a forward pass on $m = (n + t - 1)$ tokens, thus the total cost across all $T$ decoding steps is:

$$O\Big(\sum_{t=1}^{T}(n + t - 1)^2\,d_{\text{model}}\Big) \approx O\Big(T \cdot (n + T)^2\,d_{\text{model}}\Big).$$

**With KV Cache.** To build the cache, it is the same as performing a full forward pass over the prefix (length $n$) *once*:

$$O\big(n^2\,d_{\text{model}} + n\,d_{\text{model}}^2\big) \times L.$$

This stores the keys and values for all $n$ tokens in each layer. The per-step cost for each new token is:

$$O\big(n\,d_{\text{model}} + d_{\text{model}}^2\big) \times L.$$

Thus, the total cost with KV cache over $T$ new tokens is:

$$\underbrace{O\Big(n^2\,d_{\text{model}}\Big)}_{\text{initial prefix}} + \underbrace{O\Big(T \cdot \big(n\,d_{\text{model}} + d_{\text{model}}^2\big)\Big)}_{\text{decoding } T \text{ tokens}}.$$

Hence, once we have built the cache for the prefix, each new token only requires $O(n)$ operations for attention plus $O(d_{\text{model}}^2)$ for projections and feed-forward, rather than re-encoding the entire sequence each time.

## E. Additional Experiments

### E.1. Effect of Encoder Models

In our main results, we use the Contriever (Izacard et al., 2021) model, a widely adopted encoder for information retrieval tasks, as the embedding model $f_\theta(\cdot)$. To assess the impact of different embeddings, we conduct ablations with Sentence-BERT (SBert) (Reimers & Gurevych, 2019) and DeBERTa (He et al., 2020) as alternative encoders, evaluating RAG and MAPLE with 20, 60, and 100 pseudo-labeled examples. The results, presented in Table 6, show that while performance varies across encoders, MAPLE consistently outperforms the RAG baseline, demonstrating its robustness and effectiveness across different embedding choices.

### E.2. Ablation Study of Influence Score Components

We conduct an ablation study to analyze the components of the influence score defined in Equation 8, namely the geometric mean of the numbers of shortest paths $\widetilde{P_S}(\cdot, \cdot)$ and the average shortest path distance $\overline{L}_S(\cdot, \cdot)$. Experiments are performed under the setting with 20 labeled and 100 pseudo-labeled examples. Results are presented in Table 7.

Table 6: Impact of different embedding model $f_\theta(\cdot)$.

| Encoder | Method | Date | | | GoEmotion | | |
|---------|--------|------|------|------|------|------|------|
| | | $|\mathcal{D}_U^*|$=20 | $|\mathcal{D}_U^*|$=60 | $|\mathcal{D}_U^*|$=100 | $|\mathcal{D}_U^*|$=20 | $|\mathcal{D}_U^*|$=60 | $|\mathcal{D}_U^*|$=100 |
| SBert | RAG | 51.4 | 52.4 | 54.4 | 31.3 | 32.7 | 33.3 |
| | MAPLE | 52.7 | 54.0 | 55.2 | 34.7 | 36.7 | 37.3 |
| DeBERTa | RAG | 52.0 | 53.6 | 55.2 | 32.7 | 33.7 | 34.4 |
| | MAPLE | 54.4 | 55.2 | 57.6 | 37.3 | 37.2 | 39.3 |

While the length of the shortest path captures how quickly information can travel, it overlooks robustness—relying on a single path can be fragile to noise or minor data variations. On the other hand, using only the number of shortest paths captures redundancy but disregards distance; many long paths may not imply strong influence. Our influence score is designed to capture both efficiency (via short paths) and robustness (via multiple paths), resulting in more reliable and informative demonstration selection for many-shot ICL.

Table 7: Ablation study of components in the influence score

| Dataset | Banking77 | GoEmotion | GPQA |
|---------|-----------|-----------|------|
| $\overline{L}_S(\cdot,\cdot)$ | 75.3 | 37.6 | 36.4 |
| $\widetilde{P}_S(\cdot,\cdot)$ | 78.6 | 37.2 | 36.9 |
| Influence score | 80.8 | 38.1 | 37.4 |

