# OpenReview forum: "MAPLE: Many-Shot Adaptive Pseudo-Labeling for In-Context Learning"
_ICML.cc/2025/Conference — ICML 2025 poster_

### Official Review · Reviewer_fpdn · 2025-03-11

**Overall Recommendation:** 2

**Summary:**

The paper proposes is an effective framework for enhancing many-shot ICL performance in scenarios with limited labeled data. B  selecting unlabeled samples for pseudo-labeling based on their influence on labeled data and by adaptively selecting demonstrations tailored to each query, the method significantly reduces the reliance on costly labeled data while improving the in-context learning performance.

**Claims And Evidence:**

Yes.

**Essential References Not Discussed:**

No.

**Experimental Designs Or Analyses:**

Yes. I checked the experimental designs including baselines, implementations and ablations.

**Methods And Evaluation Criteria:**

Yes.

**Other Comments Or Suggestions:**

No.

**Other Strengths And Weaknesses:**

Strengths:
1. The paper is well-written and well-structured, making it easy to follow.

2. The ablation study is comprehensive, providing useful insights into the contributions of different components.

Weaknesses:
1. Practicality Concerns: The proposed method is overly complex and impractical for real-world applications. The framework requires constructing two influence graphs and selecting different demonstrations for each test query, resulting in significant computational costs. However, the performance improvements do not appear substantial enough to justify the high cost of this approach.

2. Unstable Gains from Pseudo-Labeling: As shown in Figure 3, increasing the number of pseudo-labeled samples does not consistently lead to performance improvements across most tasks. This introduces an additional challenge of tuning the pseudo-labeling budget. In practical scenarios, there is no reliable way to verify the correctness of pseudo-labels, which may lead to worse performance than a simple few-shot approach.

3. Limited Experimental Scope: The experiments are somewhat narrow in scope. The authors only evaluate the framework using the Gemini model and on relatively simple tasks. To strengthen the empirical validation, the paper should include experiments with additional models, such as LLaMA and Qwen. Moreover, the selected tasks should be more representative and general, such as those from the SuperGLUE and MATH benchmarks.

4. Lack of Discussion on Theoretical Upper Bound: The paper does not discuss the upper-bound performance of the proposed method. For instance, if many-shot data were fully annotated with golden labels, how would the proposed approach compare against a retrieval-augmented generation (RAG) baseline? A discussion on this aspect would provide a clearer perspective on the fundamental limitations of the method.

5. Comparison to More Direct Labeling Approaches: With the cost of large model inference decreasing, a straightforward alternative would be to use a sota model (like GPT4) to label all data directly. The paper does not clearly articulate the advantages of the proposed method over this simpler and more practical alternative. A thorough comparison is necessary to justify the additional complexity introduced by the framework.

**Questions For Authors:**

No.

**Relation To Broader Scientific Literature:**

The paper proposes selecting the most valuable demonstrations for many-shot pseudo-label learning.

**Theoretical Claims:**

I skimmed through it, but I didn’t carefully examine the proof.

---

> ### Author Rebuttal · Authors · 2025-04-01
>
> >**W1.** Practicality Concerns.
> >
> **Response**: Thank you for raising concerns regarding computational complexity. To address practicality, our framework incorporates strategies to improve efficiency significantly:
>
> By employing a KV cache, we reduce computational costs by fixing labeled and pseudo-labeled demonstrations across queries, allowing caching of demonstrations within the LLM prior to inference. As demonstrated in Table 2, this approach effectively enhances efficiency without loss of accuracy, making MAPLE more practical for real-world scenarios. A detailed analysis is provided in Appendix D.
>
> Additionally, the influence graph construction process is precomputed. In this way, we ensure that its computational cost does not scale with the number of queries, further enhancing efficiency and practical feasibility.
>
>
>
> >**W2.** Unstable Gains from Pseudo-Labeling.
> >
> **Response**: We would like to clarify that in our experiments, MAPLE **consistently outperforms other baselines**. MAPLE is primarily compared quantitatively against 5 baseline methods on 8 datasets with 5 settings (i.e., the number of pseudo-labeled samples). The results are shown in Figure 1. Among the 8 datasets, MAPLE performs the best on 5 datasets in all settings. On the other 3 datasets, MAPLE performs the best in 4 out of 5 settings. Therefore, these results can validate MAPLE's strong performance.
>
>
> >**W3.** Limited Experimental Scope.
> >
> **Response**: We appreciate your suggestion regarding the experiments. Our current experiments include both Gemini 1.5 Pro and Gemini 1.5 Flash, which are widely adopted and representative models for many-shot ICL [1,2]. Due to limited rebuttal time, we were unable to include additional models like LLaMA and Qwen, but we plan to explore them in future work. To broaden task diversity, we have added results on a math benchmark GSM8K, and the results demonstrate the superiority of MAPLE on math tasks. We will expand to more datasets in the next version.
>
> [1] Agarwal et al. Many-Shot In-Context Learning. NeurIPS 2024.
>
> [2] Baek et al. Revisiting In-Context Learning with Long Context Language Models. arXiv 2024.
>
> |Method|20|60|100|
> |-|-|-|-|
> |Random|90.0|91.0|91.5|
> |RAG|90.5|92.0|93.0|
> |MAPLE|92.5|94.0|95.0|
>
>
> >**W4.** Lack of Discussion on Theoretical Upper Bound: The paper does not discuss the upper-bound performance of the proposed method. For instance, if many-shot data were fully annotated with golden labels, how would the proposed approach compare against a retrieval-augmented generation (RAG) baseline? A discussion on this aspect would provide a clearer perspective on the fundamental limitations of the method.
> >
> **Response**:
>
> We appreciate your point. While deriving a theoretical upper bound for many-shot ICL is impractical due to the complexity of LLMs like Gemini 1.5 Flash, we provide an empirical upper bound by comparing MAPLE to RAG using 40, 80, and 120 fully labeled examples. MAPLE consistently outperforms RAG even under full annotation, highlighting its effectiveness and robustness beyond limited-label settings.
>
> |Embed|GPQA|Banking77|
> |-|-|-|
> |RAG+Golden|36.8/37.8/40.4|78.0/81.7/83.3|
> |MAPLE+Golden|38.3/42.4/44.9|79.3/81.7/86.2|
>
> >**W5.** Comparison to More Direct Labeling Approaches: With the cost of large model inference decreasing, a straightforward alternative would be to use a sota model (like GPT4) to label all data directly. The paper does not clearly articulate the advantages of the proposed method over this simpler and more practical alternative. A thorough comparison is necessary to justify the additional complexity introduced by the framework.
> >
> **Response**:  Thank you for raising this important point. We agree that directly using a state-of-the-art model like GPT-4 (or Gemini) to label all data is a natural and increasingly viable alternative. In fact, our zero-shot baseline involves using Gemini to label the entire dataset directly. As shown in Figure 3, MAPLE significantly outperforms this zero-shot approach across all tasks.
>
> Moreover, MAPLE only requires labeling a very small portion of the data. For instance, XSum contains over 2 million training examples, yet MAPLE achieves strong performance with at most 100 pseudo-labeled samples—representing a reduction of over 99.99% in labeling cost, even with decreasing inference costs.
>
> To further highlight MAPLE’s advantage, we include results on GPQA using Gemini 1.5 Pro. In Table 1, MAPLE achieves 44.9% accuracy, while—as reported in Figure 8 in [1]—even fully using all labeled data yields less than 44% accuracy. This clearly demonstrates the effectiveness and efficiency of our method.
>
> [1] Agarwal et al. Many-Shot In-Context Learning. NeurIPS 2024.

---

> > ### Comment · Reviewer_fpdn · 2025-04-06
> >
> > I appreciate the authors’ detailed rebuttal and the additional experiments, especially the inclusion of new results on GSM8K and comparisons under fully labeled settings.
> >
> > While the authors make a reasonable case for improved efficiency through caching and precomputing influence graphs, my primary concern around the practicality remains, i.e., the proposed method still involves multiple parts—pseudo-labeling, influence estimation, adaptive demonstration selection. In many realistic settings where simplicity and interpretability are crucial, this level of complexity may be difficult to justify, especially given the relatively modest performance gains in some scenarios.
> >
> > In light of the new evidence, I will raise my score to weak reject, acknowledging the merits of the empirical updates and clearer discussion.

---

> > > ### Author Response · Authors · 2025-04-07
> > >
> > > We thank you for your further comments and appreciate the opportunity to respond. While we understand your concerns regarding practicality and perceived complexity, we would like to emphasize that the **performance improvements are meaningful**, and **the modular design does not render the method impractical**.  Our clarifications are as follows:
> > >
> > > 1. **Performance improvements are meaningful.** To demonstrate simplicity and interpretability, we directly compare MAPLE with the zero-shot labeling baseline, which you've described in Weakness 5 as “more practical and straightforward” (i.e., using a SoTA model to label all data directly). We report average performance across **different task types** and find that **MAPLE significantly outperforms the practical baseline**, particularly on classification tasks, where it achieves a relative improvement of 50%. These substantial gains underscore that, **despite involving multiple components, MAPLE delivers performance improvements that justify the added complexity.**
> > >
> > > |Method|Summarization|Reasoning|Classification|Question Answering|
> > > |-|-|-|-|-|
> > > |Zero-shot|16.1|49.5|42.5|34.3|
> > > |MAPLE|20.1|53.7|66.9|37.7|
> > >
> > > 2. **The modular design does not render the method impractical**.
> > >
> > > - First we would like to claim that **each component in MAPLE contributes meaningfully to the final performance**. We provide detailed evidence through breakdowns and ablation studies, as referenced in our responses to Reviewer K6CP (W2) and Reviewer qNQK (Q3).
> > > - Second, the computational cost of MAPLE remains manageable.
> > >   - (1) **Pseudo-labeling**: we enhance efficiency by **selectively** pseudo-labeling only the top-P nodes with the highest influence scores, instead of pseudo-labeling all train data, significantly reducing API calls.
> > >   - (2) **Influence estimation**: The graph construction requires the computation of the relevance score $r$ among any pair of nodes, which will be $\mathcal{O}(|\mathcal{V}|^2)$. To compute shortest paths, we use breadth-first search for each node, and the cost is $\mathcal{O}(|\mathcal{V}|+|\mathcal{E}|) = \mathcal{O}(|\mathcal{V}|)$ as $\mathcal{E}=\mathcal{O}(k|\mathcal{V}|)$. Therefore, the whole shortest path computation cost is $\mathcal{O}(|\mathcal{D}_L||\mathcal{V}|) = \mathcal{O}(|\mathcal{V}|)$ as $|\mathcal{D}_L|\ll |\mathcal{D}|=|\mathcal{V}|$. Notably, the above cost is only required **once** before inference, and **does not scale with the number of test-time queries**. With more queries involved during the test, the computational cost of the graph becomes more negligible.
> > >   - (3) **Adaptive demonstration selection**: We emphasize that adaptive demonstration selection is an **optional component** that offers a **trade-off between efficiency and performance**, as discussed in Sec. 4.5. In MAPLE, we incorporate personalized demonstrations for each query, which incurs additional cost but effectively filters out unhelpful examples, leading to better performance. As shown in Figure 3, this adaptive strategy also improves performance in RAG settings. To accommodate efficiency-focused scenarios, we also provide a variant of MAPLE with fixed demonstration selection and KV caching (Sec. 4.5). This variant enables faster inference with only a mild sacrifice in performance, offering a flexible solution based on deployment needs. The complexity comparison of MAPLE and the KV cache variant is provided in Appendix D.
> > >
> > >
> > > In summary, **each component of MAPLE is either lightweight or designed to offer a meaningful trade-off between performance and efficiency**. We also provide practical variants to accommodate different deployment scenarios. Therefore, we sincerely hope that our detailed responses can help clarify the practical aspects of our framework and address your concerns. Thank you so much for your effort in reviewing our work!
> > >
> > > Sincerely,
> > > Authors of Submission 14586

---

### Official Review · Reviewer_y2DN · 2025-03-12

**Overall Recommendation:** 4

**Summary:**

This paper presents MAPLE, a method for pseudo-labeling in many-shot ICL settings. Key innovation includes similarity-based selection for pseudo-labeling and demonstration example selection.

**Claims And Evidence:**

1.	It’s interesting to study many-shot ICL under pseudo-label settings, which has practical value.
2.	The author claims “strong performance” for the MAPLE, but didn’t specify the baselines nor quantitative results for comparison.

**Essential References Not Discussed:**

NA

**Experimental Designs Or Analyses:**

See methods

**Methods And Evaluation Criteria:**

1.	The baselines are reasonable, and the datasets are up-to-date and commonly used.
2.	It seems the embedding module is important for MAPLE, and an ablation on that is important.

**Other Comments Or Suggestions:**

1.	**IMPORTANT** Most figures are not rendered correctly. Labels for legend and axes are missing.

**Other Strengths And Weaknesses:**

NA

**Questions For Authors:**

1.	Do you have any explanation why MAPLE works well for GPQA, given the diversity of topics in GPQA?

**Relation To Broader Scientific Literature:**

It's an interesting extension towards both many-shot ICL and pseudo-labeling.

**Theoretical Claims:**

NA

---

> ### Author Rebuttal · Authors · 2025-04-01
>
> >**Claim** The author claims “strong performance” for the MAPLE, but didn’t specify the baselines nor quantitative results for comparison.
> >
> **Response**: In our experiments, MAPLE is primarily compared quantitatively against 5 baseline methods on 8 datasets with 5 settings (i.e., the number of pseudo-labeled samples). The results are shown in Figure 3. Among the 8 datasets, MAPLE performs the best on 5 datasets on all settings. On the other 3 datasets, MAPLE performs the best on 4 out of 5 settings. Therefore, these results can validate MAPLE's strong performance.
>
>
> >**Exp** It seems the embedding module is important for MAPLE, and an ablation on that is important.
> >
> **Response**: Thank you for the suggestion. We have conducted ablations using Sentence-BERT (SBert) [1] and DeBERTa [2] as alternative embedding models, evaluating MAPLE with 20, 60, and 100 pseudo-labeled examples. While performance varies across models, MAPLE consistently outperforms baseline, demonstrating its robustness and effectiveness regardless of the specific embedding choice.
>
> |Embed|Date|GoEmotion|
> |-|-|-|
> |RAG+SBert|51.4/52.4/54.4|31.3/32.7/33.3|
> |MAPLE+SBert|52.7/54.0/55.2|34.7/36.7/37.3|
> |RAG+DeBERTa|52.0/53.6/55.2|32.7/33.7/34.4|
> |MAPLE+DeBERTa|54.4/55.2/57.6|37.3/37.2/39.3|
>
> [1] Reimers N, Gurevych I. Sentence-BERT: Sentence Embeddings using Siamese BERT-Networks.
>
> [2] He P, Liu X, et al. DeBERTa: Decoding-enhanced BERT with Disentangled Attention.
>
> >**Comm1.** IMPORTANT Most figures are not rendered correctly. Labels for legend and axes are missing.
> >
> **Response**: We sincerely appreciate your feedback on figure clarity. In the revision, we will ensure all figures are correctly formatted and rendered.
>
> >**Q1.** Do you have any explanation why MAPLE works well for GPQA, given the diversity of topics in GPQA?
> >
> **Response**: Thank you for pointing this out. MAPLE's strong performance on GPQA can be attributed to its adaptive demonstration selection, which tailors pseudo-labeled demonstrations specifically for each test query. This adaptability allows MAPLE to effectively handle the topic diversity in GPQA by selecting demonstrations that are contextually relevant to each individual query. Consequently, MAPLE can leverage pseudo-labeled samples to improve performance even when the topics are diverse.

---

> > ### Comment · Reviewer_y2DN · 2025-04-04
> >
> > Thanks for the response. You can consider putting the abalation results in the Appendix. I don't have any other comment, and I'll keep my current rating.

---

> > > ### Author Response · Authors · 2025-04-04
> > >
> > > Dear Reviewer y2DN,
> > >
> > > Thank you for your thoughtful review and constructive feedback. We appreciate your time and effort, and to strengthen the quality of our work, we will include the ablation results in the Appendix in the final version.
> > >
> > > Best regards,
> > >
> > > Authors

---

### Official Review · Reviewer_qNQK · 2025-03-12

**Overall Recommendation:** 3

**Summary:**

The paper considers semi-supervised many-shot in-context learning setting, i.e., having small labeled and large unlabeled support sets to perform in-context learning with long-context LLMs. Authors argue that within this problem setting it would be beneficial to (i) identify the most impactful unlabeled samples to pseudo-label them, and, subsequently (ii) use adaptive example selection mechanism to select examples for each test query from the union of the pseudo-labeled examples along with labeled ones. To reach both, authors leverage the concept of node influences in graphs. In particular, first, the graph is built with nodes representing examples from both labeled and unlabeled sets, and edges are assigned according to the similarity of the examples in some embedding space. Subsequently, top-p nodes from the set of unlabeled examples are selected according to the score that is lower bound to the node influence. Adaptive demonstration selection is built in the similar fashion, relying on node influence of the labeled and pseudo-labeled samples on the test query. Authors evaluate their approach of the diverse set of problems and show that it outperforms the considered baselines.

**Claims And Evidence:**

See Questions for Authors.

**Essential References Not Discussed:**

N/A

**Experimental Designs Or Analyses:**

Overall, seems valid.

**Methods And Evaluation Criteria:**

The proposed method is evaluated using challenging datasets and with the recent models.

**Other Comments Or Suggestions:**

N/A

**Other Strengths And Weaknesses:**

See Questions for Authors.

**Questions For Authors:**

1. Do you really need to cut Top-K edges for each node? Since, you still will be selecting top-P nodes only after all, and it’s single cost thing, you can just run Top-P selection over fully connected graph? In other words, is it correct that Top-K only controls speed and does not affect the performance, or there is some interplay between both Top-K and Top-P that affect the performance?

2. Given Figure 6 shows that, mostly, it is more beneficial to put pseudo-labeled examples at back and examples with ground truth labels closer to the query, could you also run the baseline when you just put questions for unlabeled samples (without pseudolabels). It is basically Unsupervised ICL of [1], which was shown to improve upon few-shot baseline that employs only labeled set.

3. What I am currently missing is the ablation of what part of the proposed approach is actually the most important or the evidence that both are important.

4. There are some inconsistencies in set of baselines for different tables and Figures. For example, Table 3 misses zero-shot, few-shot and RAG-Adapt (compared to Figure 1). Similarly, Figure 4 misses RAG-Adapt.

5. Maybe I missed it somewhere, but what was used as the embedding model $f$ to construct graphs?

6. Can we compute node influence directly and not rely on the lower-bound? Does lower bound have some benefits? How the performance will be different if the node influence would be computed directly?

[1] Agarwal et al. Many-Shot In-Context Learning. NeurIPS 2024.

**Relation To Broader Scientific Literature:**

Many-shot in-context learning is very recent and promising approach to perform adaptation of large-context LLMs. Given the high data labeling cost, it is important to consider the semi-supervised setting, thus I believe that the paper studies an important topic.

**Theoretical Claims:**

I checked Theorem 3.2 and briefly checked the proof of Lemma A.1.

---

> ### Author Rebuttal · Authors · 2025-04-01
>
> >**Q1.** Do you really need to cut Top-K edges for each node? Is it correct that Top-K only controls speed and does not affect the performance, or there is some interplay between both Top-K and Top-P that affect the performance?
> >
> **Response**: Thank you for the question. We want to clarify that Top-K pruning is essential and does influence selection. Our influence score in Eq(8) depends on the shortest path and its count. In a fully connected graph, all node pairs are directly connected with only one shortest path of length 1, which removes meaningful structural differences and reduces influence estimation to near-random. Moreover, computing the shortest path on a fully connected graph incurs complexity in $\mathcal{O}(|\mathcal{V}|^2)$. Thus, Top-K is important for both efficiency and effective selection.
>
> >**Q2.** Given Figure 6 shows that, mostly, it is more beneficial to put pseudo-labeled examples at back and examples with ground truth labels closer to the query, could you also run the baseline when you just put questions for unlabeled samples (without pseudolabels). It is basically Unsupervised ICL of [1], which was shown to improve upon few-shot baseline that employs only labeled set.
> >
> **Response**: Thank you for the suggestion, and we have added results using only unlabeled questions without pseudo-labels, following the unsupervised ICL setup (#labeled=20, #unlabled=100). Consistent with Figure 6, we observe that placing labeled examples closer to the query still leads to better performance. However, without pseudo-labels, the benefit from unlabeled examples is limited (compared to Fig 6.), highlighting the importance of label information—even if approximated—for effective many-shot ICL.
> |Dataset|Salient|GPQA|
> |-|-|-|
> Rand w/w.o. R|65.6/66.0|33.3/34.3|
> Rag w/w.o. R|68.8/68.2|33.8/34.8|
> MAPLE w/w.o. R|70.4/70.8|35.3/36.1|
>
> >**Q3.** What part of the proposed approach is actually the most important?
> >
> **Response**: We appreciate the reviewer’s question. The RAG-Adapt baseline in our paper can be seen as a variant of MAPLE without the graph structure, as both rely on Contriever for relevance score. Without pseudo-labeling, as in our response to Q2, performance drops due to the lack of label information. Further, in our response to Reviewer K6CP W2, we detail how removing individual components of the influence score degrades performance. Together, these results demonstrate that each part of MAPLE is crucial to its overall effectiveness.
>
> >**Q4.** There are some inconsistencies in set of baselines for different tables and Figures. For example, Table 3 misses zero-shot, few-shot and RAG-Adapt (compared to Figure 1). Similarly, Figure 4 misses RAG-Adapt.
> >
> **Response**: Thank you for pointing this out. We believe the reference is to Table 1 (rather than Table 3) since Table 3 is a list of prompts. Due to rebuttal time constraints and API budget limits, we have now additionally run Gemini 1.5 Flash for zero-shot, few-shot, and RAG-Adapt on Table 1, as well as RAG-Adapt for Figure 4. We will include these updated results in the revised version of the paper.
>
> |Datset|0-shot|few-shot|20|40|60|80|100|
> |-|-|-|-|-|-|-|-|
> |Banking77|75.1|76.9|77.0|76.9|76.5|76.7|78.5|
> |Date|49.1|52.9|53.6|53.5|53.1|55.4|56.1|
> |GPQA|34.3|35.8|37.1|36.2|35.7|32.5|34.0|
>
> |Datset|50|100|150|200|
> |-|-|-|-|-|
> |Banking77|79.3|81.3|81.3|81.7|
> |Date|56.4|59.2|59.4|63.4|
>
> >**Q5.**What was used as the embedding model to construct graphs?
> >
> **Response**: Thank you for the question. We use Contriever as the embedding model. This is mentioned in the right column of line 115: “...the relevance score r(vi, vj), as defined in Contriever...” For clarity, we will also explicitly restate this in the Implementation section in the revised version.
>
> >**Q6.** Can we compute node influence directly and not rely on the lower-bound? Does lower bound have some benefits? How the performance will be different if the node influence would be computed directly?
> >
> **Response**: We claim that it is computationally impractical to directly calculate the node influence. As stated in Eq. (17) in Appendix A, the node influence between any pair of nodes, $v_i$ and $v_j$, is calculated from the iterative expansion of the neighboring nodes of $v_i$. This involves all the nodes that exist in any path between $v_i$ and $v_j$, and their corresponding derivatives regarding the embedding of $v_j$. This can be computationally prohibitive when the number of such nodes is massive due to the long distance between $v_i$ and $v_j$.
>
> Therefore, we propose to rely on the lower bound to compute the influence score instead of directly computing the node influence. With our proposed Theorem 3.2, the influence score is computed based on the shortest path distance and the number of shortest paths. Thus, it is much easier to compute, compared to the massive computation of derivatives in the original node influence.

---

### Official Review · Reviewer_K6CP · 2025-03-16

**Overall Recommendation:** 3

**Summary:**

This work develops a semi-supervised in-context learning framework by exploiting small amount of labeled and large unlabelled dataset. A Ken graph is built upon the labeled and unlabelled dataset. The unlabelled samples (nodes) that are similar to the labeled ones are selected
For pseudo labelling. Finally, demonstrations that are highly relevant to the test query are selected from the combined dataset for prediction.

**Claims And Evidence:**

The claims made in this paper are mainly supported with empirical results.

**Essential References Not Discussed:**

N/A

**Experimental Designs Or Analyses:**

The experiment designs are mostly appropriate.  However, evaluation on higher number of pseudo labeled samples are missing which may demonstrate the potential upperbound of the proposed method.

**Methods And Evaluation Criteria:**

The methodology and benchmarking datasets are mostly appropriate.

**Other Comments Or Suggestions:**

It is unclear why the x axis starts with 20 in figure 3. With 0 pseudo-labeled samples, would this be equivalent to few-shot?

**Other Strengths And Weaknesses:**

Strength:
1. Exploiting graph for relevance calculation could exploit the data manifold and potentially leads to better results.

2. Semi-supervised in-context learning can alleviate the reliance on excessive human annotation.

Weakness:
1. Building a graph for selecting relevant unlabelled samples itself induces additional computation overhead. There is no discussions on the computation cost for graph construction.

2. The design for influence score seems arbitrary. It would be good to see if keeping only the shortest path or number of shortest paths is worse.

3. It is worth noting that the impact of increasing the number of pseudo labeled samples is indeterministic. For GoEmotion, Banking77 and Date, the performance may still go up with more pseudo-labels. While, Tracking 7 seems does not benefit any pseudo labeled samples. A deeper analysis is necessary.7

**Questions For Authors:**

- Please further explain why increasing the number of pseudo labels may harm certain datasets.

**Relation To Broader Scientific Literature:**

Semi-supervised learning has been thoroughly investigated. The proposed method exploits a graph and the influence score shares similarity with graph based SSL and label propgation. However, integrating SSL to ICL is novel.

**Theoretical Claims:**

Not thoroughly checked.

---

> ### Author Rebuttal · Authors · 2025-04-01
>
> >**W1.** Building a graph for selecting relevant unlabelled samples itself induces additional computation overhead. There is no discussions on the computation cost for graph construction.
> >
> **Response**: Thank you for bringing up this point. The graph construction requires the computation of the relevance score $r$ among any pair of nodes, which will be $\mathcal{O}(|\mathcal{V}|^2)$. To compute shortest paths, we use breadth-first search for each node, and the cost is $\mathcal{O}(|\mathcal{V}|+|\mathcal{E}|) = \mathcal{O}(|\mathcal{V}|)$ as $\mathcal{E}=\mathcal{O}(k|\mathcal{V}|)$. Therefore, the whole shortest path computation cost is $\mathcal{O}(|\mathcal{D}_L||\mathcal{V}|)$. Notably, the above cost is only required **once** before inference and thus is independent of the number of queries. With more queries involved during the test, the computational cost of the graph becomes more negligible. Moreover, as shown in Table 2 in our paper, the adaptive demonstration selection component does not incur much computational cost.
>
> Thank you so much for your suggestion. We will include this discussion in the appendix.
>
> >**W2.** The design for the influence score seems arbitrary. It would be good to see if keeping only the shortest path or number of shortest paths is worse.
> >
> **Response**: We appreciate your suggestion and have added results (#labeled=20, #p-labeled=100) using only the shortest path and the number of shortest paths. While the shortest path captures how quickly information can travel, it overlooks robustness—relying on a single path can be fragile to noise or minor data variations. On the other hand, using only the number of shortest path captures redundancy but disregards distance; many long paths may not imply a strong influence. Our influence score is designed to capture both efficiency (via short paths) and robustness (via multiple paths), resulting in a more reliable and informative demonstration selection for many-shot ICL.
>
>
> |Dataset|Banking77|GoEmotion|GPQA|
> |-|-|-|-|
> |len(shortest path)|75.3|37.6|36.4|
> |# \|shortest path\||78.6|37.2|36.9|
> |Influence score|80.8|38.1 |37.4|
>
> >**W3.** It is worth noting that the impact of increasing the number of pseudo labeled samples is indeterministic. For GoEmotion, Banking77 and Date, the performance may still go up with more pseudo-labels. While, Tracking 7 seems does not benefit any pseudo labeled samples. A deeper analysis is necessary.
> **Q1.** Please further explain why increasing the number of pseudo labels may harm certain datasets.
> >
> **Response**:
> Thank you for highlighting this point. The inclusion of additional pseudo-labeled samples can sometimes harm performance because pseudo-labels generated by LLMs are not always accurate. Incorrect pseudo-labels may introduce misleading information when used as demonstrations, negatively influencing LLM predictions for specific datasets [1]. However, our method addresses this issue by adaptively selecting pseudo-labeled samples based on their relevance to each test query. This approach mitigates the negative impact of inaccurate pseudo-labeling, as demonstrated by consistent improvements across most tasks. We note that the performance decline is limited to certain datasets, likely due to the higher difficulty or inherent ambiguity of the samples. Nevertheless, MAPLE consistently outperforms baselines even though the number of pseudo-labeled samples is not optimal.
>
> [1] Agarwal et al. Many-Shot In-Context Learning. NeurIPS 2024.
>
> >**Comm1.** It is unclear why the x axis starts with 20 in figure 3. With 0 pseudo-labeled samples, would this be equivalent to few-shot?
> >
> **Response**: Thank you for the observation. Yes, when the number of pseudo-labeled samples is 0, Random, RAG, and MAPLE all reduce to the few-shot setting. To avoid redundancy, we omit x=0 from the x-axis and instead include the few-shot performance as a green horizontal dashed line in Figure 3 for comparison. We will clarify this explicitly in the revised version.

---

### Decision · Program_Chairs · 2025-05-01

**Decision:**

Accept (poster)

**Comment:**

This paper proposes a semi-supervised many-shot in-context learning approach - having small labeled and large unlabeled support sets to perform in-context learning with long-context LLMs.

Strengths:

- Very important line of work and has potential to be used across various academic and industrial applications.

- Solid experiments and results.

I enjoyed reading this work, and authors provide a detailed rebuttal addressing most of the concerns raised by reviewers. Specifically, major concerns about computation overhead for graph construction and ablation of different components of MAPLE are addressed by authors in the rebuttal. Overall, I recommend accept.